# An on-demand, drop-on-drop method for studying enzyme catalysis by serial crystallography

Agata Butryn [1,2,17], Philipp S. Simon [3,17], Pierre Aller [1,2,17], Philip Hinchliffe[4], Ramzi N. Massad[3], Gabriel Leen[5,6], Catherine L. Tooke[4], Isabel Bogacz[3], In-Sik Kim[3], Asmit Bhowmick[3], Aaron S. Brewster [3], Nicholas E. Devenish[1], Jürgen Brem [7], Jos J. A. G. Kamps [7,13], Pauline A. Lang [7], Patrick Rabe[7], Danny Axford [1], John H. Beale[1,14], Bradley Davy[1,15], Ali Ebrahim[1], Julien Orlans [1,8], Selina L. S. Storm [1,16], Tiankun Zhou[1,2], Shigeki Owada[9,10], Rie Tanaka[9,11], Kensuke Tono[9,10], Gwyndaf Evans [1], Robin L. Owen [1], Frances A. Houle[12], Nicholas K. Sauter [3], Christopher J. Schofield [7], James Spencer [4], Vittal K. Yachandra[3], Junko Yano [3], Jan F. Kern [3✉] & Allen M. Orville [1,2✉]

Serial femtosecond crystallography has opened up many new opportunities in structural biology. In recent years, several approaches employing light-inducible systems have emerged to enable time-resolved experiments that reveal protein dynamics at high atomic and temporal resolutions. However, very few enzymes are light-dependent, whereas macromolecules requiring ligand diffusion into an active site are ubiquitous. In this work we present a drop-on-drop sample delivery system that enables the study of enzyme-catalyzed reactions in microcrystal slurries. The system delivers ligand solutions in bursts of multiple picoliter-sized drops on top of a larger crystal-containing drop inducing turbulent mixing and transports the mixture to the X-ray interaction region with temporal resolution. We demonstrate mixing using fluorescent dyes, numerical simulations and time-resolved serial femtosecond crystallography, which show rapid ligand diffusion through microdroplets. The drop-on-drop method has the potential to be widely applicable to serial crystallography studies, particularly of enzyme reactions with small molecule substrates.

[1] Diamond Light Source, Harwell Science and Innovation Campus, Didcot, UK. [2] Research Complex at Harwell, Rutherford Appleton Laboratory, Didcot, UK. [3] Molecular Biophysics and Integrated Bioimaging Division, Lawrence Berkeley National Laboratory, Berkeley, CA, USA. [4] School of Cellular and Molecular Medicine, University of Bristol, University Walk, Bristol, UK. [5] PolyPico Technologies Ltd, Unit 10, Airways Technology Park, Rathmacullig West, Cork, Ireland. [6] Department of Electronic and Computer Engineering, University of Limerick, Limerick, Ireland. [7] Department of Chemistry, Chemistry Research Laboratory, University of Oxford, Oxford, UK. [8] UMR0203, Biologie Fonctionnelle, Insectes et Interactions, Institut National des Sciences Appliquées de Lyon, Institut National de Recherche pour l'Agriculture, l'Alimentation et l'Environnement, University of Lyon, Villeurbanne, France. [9] RIKEN SPring-8 Center, Hyogo, Japan. [10] Japan Synchrotron Radiation Research Institute, Hyogo, Japan. [11] Department of Cell Biology, Graduate School of Medicine, Kyoto University, Kyoto, Japan. [12] Chemical Sciences Division, Lawrence Berkeley National Laboratory, Berkeley, CA, USA. [13] Present address: Diamond Light Source, Harwell Science and Innovation Campus, Didcot, UK. [14] Present address: Paul Scherrer Institut, Villigen PSI, Switzerland. [15] Present address: School of Computing, University of Leeds, Leeds, UK. [16] Present address: European Molecular Biology Laboratory, Hamburg Outstation c/o DESY, Hamburg, Germany. [17] These authors contributed equally: Agata Butryn, Philipp S. Simon, Pierre Aller. ✉email: jfkern@lbl.gov; allen.orville@diamond.ac.uk

Serial femtosecond crystallography (SFX) techniques are relatively new and are undergoing rapid development[1]. Synchronization of several femtoseconds long X-ray free-electron laser (XFEL) pulses with visible light pump lasers has been applied to time-resolved SFX (tr-SFX) studies on several systems over sub-picosecond or longer timescales[2]. Most biological systems, however, are not light-dependent, necessitating the development of methods that could broaden the applicability of tr-SFX methods to a larger variety of biological samples. While some systems can be photosensitized by the generation of light-sensitive caged substrates[3] or post-translational modification of the protein itself[4–6], these strategies are sample-specific and do not offer a routine, general solution.

Studying enzymatic interactions by tr-SFX at conditions similar to physiological is emerging as a method that helps to advance our fundamental understanding of enzyme catalysis and offers great opportunities to improve the efficiency of therapeutic strategies[7,8]. For enzyme-catalyzed reactions, triggering can be achieved by mixing protein crystals with substrate and collecting diffraction data after different time delays. Theoretical calculations predict that, using appropriately small crystals (<5 μm), millisecond diffusion times are achievable[7–9], which is sufficiently faster than many enzyme-catalyzed reactions (~70 ms) and should enable the substrate to occupy the majority of active sites before significant reaction occurs[2,7–9]. Based on these findings, a number of approaches to investigate enzyme reactions are currently being pursued, both at XFELs and synchrotron X-ray sources. These have progressed rapidly in recent years to cover time delays from multisecond[10–13] to sub-second timescales[14–16]. Mixing approaches are, however, not easy to implement as they require rapid, reliable, and controlled mixing without damaging the crystal lattice[9]. In this context, time-resolved serial synchrotron crystallography (tr-SSX) experiments can be particularly challenging, since much longer X-ray exposure times (e.g., ~ 1 ms to several tens of microseconds) are convoluted with reaction initiation strategies that together limit the overall time resolution achievable for the reaction under investigation.

The most established technique used in mixing experiments involves the sample being presented to the X-ray beam in the form of a liquid jet, which is characterized by a considerably high sample and ligand flow rate(s) and is therefore not suited to samples with limited availability[11–13,16]. Because of the large volumes of microcrystal slurry and ligand solutions demanded by continuous flow, sample consumption is of particular concern, especially since tr-SFX/SSX experiments typically need to cover a broad range of time delays necessary to create a 'molecular movie' across the reaction cycle. Recent promising designs include samples being deposited onto a moving tape as a continuous stream[10] and the liquid application method for time-resolved studies (LAMA), where substrate solution is applied from a piezoelectric dispenser onto a chip preloaded with microcrystals[15].

Here, we show an alternative on-demand droplet-based mixing strategy for time-resolved mixing experiments. This system builds on the drop-on-tape method, which has been shown to retain fidelity of crystalline order, provides low sample consumption, high hit rates and, above all, very high levels of versatility[14,17–21]. We modified the drop-on-tape design to accommodate two droplet dispensing heads able to introduce a burst of a variable number of picoliter-size drops of concentrated ligand solutions at 1–30 kHz dispensing frequency on a previously dispensed larger, nanoliter-size crystal-containing drop (drop-on-drop). The system is very flexible, with Kapton tape speeds ranging from 600 to 10 mm s$^{-1}$ (0.1–6 s mixing time) and can initiate enzyme reactions by adding between 1 and 300 drops of substrate (≈30–200 pL each) to the main crystal slurry drop. To investigate the potential of the drop-on-drop method, we performed numerical simulations and proof-of-principle mixing experiments using fluorescent dyes. We also used hen egg white lysozyme (HEWL) and a serine β-lactamase (SBL), CTX-M-15, as two case studies to demonstrate the application of this method for enzyme-catalyzed reactions in crystals. Our results show that the drop-on-drop mixing strategy is capable of achieving sub-second time resolution with dramatically reduced ligand consumption and should be therefore readily applicable to explorations of a wide range of enzyme-substrate systems, including those such as the SBLs that are relevant to health and active or potential drug targets.

## Results

**Drop-on-drop experimental setup enables accelerated mixing in droplets.** In the drop-on-drop mixing system design, substrate drops are added with piezoelectric injector (PEI), each carrying ~120 pL, by collision at a relative velocity of 1–2 m s$^{-1}$ with the main, crystal-bearing ADE drop. The drops are transported (30–300 mm s$^{-1}$) by the Kapton tape to the interaction region, which for SFX experiments was the location of the X-ray beam at SACLA (Fig. 1a, Supplementary Fig. 1, Supplementary Movie 1 and 2). To investigate whether collision-driven mass flow caused by the substrate drops arriving with a velocity of 1–2 m s$^{-1}$ and at high frequency can accelerate mixing when compared to diffusion only, we carried out two sets of simulations on calcium-sensitive fluorescent dyes (see the "Methods" section, Supplementary Note 1, Supplementary Figs. 2–4, Supplementary Tables 1–3). In a three-dimensional reaction–diffusion system, the diffusion and binding of the calcium ligand from stationary picoliter droplets through a nanoliter-sized droplet containing fluorescent dyes takes considerably longer than the average enzymatic reaction (1–2 s, Fig. 1b). The reaction-mass-transfer simulation revealed, however, that the high-frequency and high-velocity collision of several small droplets distributed across the surface of the larger drop will decrease the equilibration time to <1 ms and to a point where the overall drop size and diffusion are less restrictive. To test how much the mixing is accelerated in the drop-on-drop approach, we performed proof-of-principle experiments using the same calcium-sensitive fluorescent dye system and compared these results with the simulations (Fig. 1b, Supplementary Note 1, Supplementary Figs. 5 and 6). The observations show that the rise of the experimentally measured fluorescence components, with rise times of <100 to 150 ms, is notably faster than that expected based on the simulated diffusion data (one order of magnitude). This confirmed that the mixing process induced by a substrate droplet colliding with the enzyme-carrying drop is accelerated relative to simple diffusion, to a point where saturating the enzymatic active sites faster than the catalytic reaction progresses becomes possible.

**Drop-on-drop enables tracking enzyme-catalyzed reactions with serial X-ray crystallography.** To demonstrate the application of the drop-on-drop method to enzyme-catalyzed reactions in crystals, we used two different enzyme systems as case studies: hen egg white lysozyme (HEWL) and a bacterial serine β-lactamase (SBL), CTX-M-15. HEWL is a widely used model system for crystallographic methods development[10,15], while CTX-M-15 is a class A, extended spectrum SBL that is distributed worldwide in multiple bacterial pathogens where it is responsible for resistance to a range of β-lactam antibiotics with the exception of carbapenems[22]. The basis for the poor turnover of these last-resort β-lactam antibiotics (for example, ertapenem) and the mechanistic details of inhibition are currently not well understood, largely due to the lack of structural information. Using the drop-on-drop system at the SACLA XFEL source, we first obtained a series of time-resolved HEWL structures recorded at

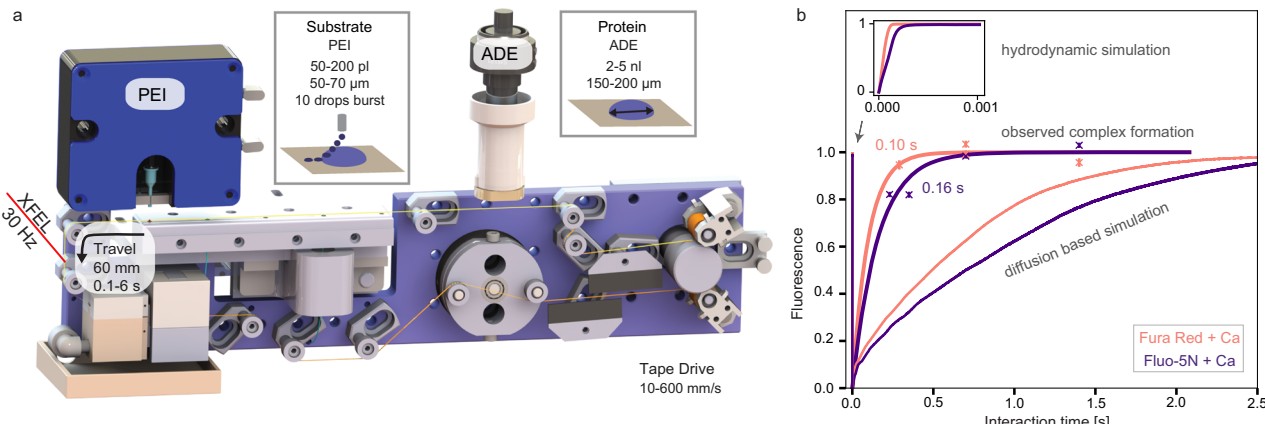

**Fig. 1 Drop-on-drop experimental setup enables accelerated mixing in droplets. a** Overview of the setup. The protein crystals are ejected by an acoustic droplet ejector (ADE, white and dark gray column above tape and on the right-hand side) and synchronized to the XFEL master clock. The burst of substrate drops is ejected by a piezoelectric injector (PEI, blue box above tape, on the left-hand side) that can accommodate disposable cartridges with a range of orifices that dictate the size of the droplets. Reaction time is varied by the speed of the tape drive (10–600 mm s$^{-1}$) and depends upon the location of the PEI with respect to the ADE droplet intersection and the X-ray interaction points; in the case reported here, we selected time points between 0.1 and 6 s. **b** Relative fluorescence intensity (crosses) of the calcium-bound forms of Fura Red (salmon) and Fluo-5N (purple) dyes obtained from the scaling of the normalized spectral components and their errors of the fits (Supplementary Fig. 5) compared to reaction simulations of collision-driven hydrodynamic flow (inset) and equilibration by diffusion. Lines correspond to the fitted exponential functions of the experimental data and rise times are indicated. Fluorescence signal of Fura Red rises faster than Fluo-5N due to the lower dissociation constant.

0.2, 0.6, and 2 s after addition of two to six drops of an inhibitor, N-acetyl-D-glucosamine (GlcNAc, 16.7–43.7 mM final concentration, Supplementary Tables 4 and 5). As expected, the resting state control model shows an active site with ordered solvent molecules but devoid of organic ligands. Although the structure determined at the 0.2 s time point is essentially identical to the resting state model, the isomorphous difference and polder OMIT electron density maps for the 0.6 and 2 s delay times display strong evidence for the presence of a ligand in the active site despite ligand affinity in the mM range[23] (Fig. 2a, Supplementary Fig. 7). The ligand can be unambiguously assigned as GlcNAc, in a position corresponding well to that reported in other studies[24–26] (Fig. 2b–d, Supplementary Note 2).

Based upon the limited set of conditions tested at SACLA, the HEWL results clearly demonstrate that the drop-on-drop mixing method can be used for mix-and-diffuse studies of ligand binding to protein crystals on timescales down to at least 0.6 s after ligand addition to the protein microcrystal suspension. This encouraged us to collect a similar time-resolved series of structures for CTX-M-15 and a β-lactam antibiotic, ertapenem, with 0.6 and 2 s time delays between the drop-on-drop ligand addition and the X-ray exposure (Supplementary Tables 4 and 6). Although clear changes in the active site of the HEWL structure are visible after 0.6 s, no apparent changes in the active site of CTX-M-15 are present at the same time point after addition of four drops of ertapenem (110.3 mM final concentration, Fig. 2e, Supplementary Figs. 8 and 9, Supplementary Table 7). This is despite the similar crystal dimensions and crystal solvent content of the two systems (Supplementary Fig. 10), a greater molar excess of the ligand used with CTX-M-15 and higher protein-ligand affinity (11.6 μM for CTX-M-15-ertapenem [Supplementary Table 8] vs. 47.6 mM for HEWL-GlcNAc[23]). Successful binding of ertapenem in the 2 s time point data is apparent in the $F_{o\,(2\,s)} - F_{o\,(resting)}$ isomorphous difference map, since clear difference density adjacent to the nucleophilic Ser70 appears (Supplementary Fig. 9), which allows modeling of an acyl-enzyme complex into the electron density (i.e., with the β-lactam ring open by reaction with the nucleophilic serine, Fig. 2f, Supplementary Figs. 11 and 12). This was further confirmed by calculation of polder OMIT maps

which show electron density consistent with ring-opened ertapenem bound similarly to that in a control presoaked dataset (Fig. 2g, Supplementary Note 3).

## Discussion

In conclusion, the data presented here demonstrate that the on-demand, drop-on-drop method developed from the drop-on-tape XFEL sample delivery system, can indeed overcome the limitations associated with ligand diffusion through droplets. Our successful observation of ligand complex formation in our tr-SFX data at timescales below 0.6 s shows that the drop-on-drop method enables generation of enzyme complexes with small molecule ligands/substrates on timescales similar to the kinetics of enzyme-catalyzed reactions. The versatility and efficiency of drop-on-drop sample delivery substantially expands the currently very limited set of low sample consumption delivery systems able to achieve reaction times suitable for studying enzymatic reactions. Indeed, our six structures required only 84–258 μL of ligand solutions, and 162–420 μL of microcrystal slurry. For example, the 0.6 s tr-SFX HEWL dataset consumed a total of only 0.7 μmol (9.5 mg) protein and 0.18 mL of 0.226 M ligand (41 μmol total). In addition, by adding humidity and temperature control and varying the position of the PEI, the system can offer easy access to a wider range of delay times (from 50 ms to 10 s) relevant for many enzymatic reactions. Also, a large range of crystal sizes (5–100 μm) can be accommodated by this system, as the droplet size can be adjusted, and no limiting orifice is used for generation of the crystal-containing droplets. This allows to choose optimum conditions for each individual protein system, balancing needed diffraction signal strength (often increased with larger crystal size) and desired minimum diffusion equilibration time (decreased with smaller crystal size, also highly dependent on buffer viscosity).

It is important to realize how ready access to such technologies will make a major contribution to exploiting the potential of tr-SFX/SSX to transform our understanding of protein-ligand interactions and enzyme catalysis. Drop-on-drop technology can be in principle easily transferred from the XFEL to the

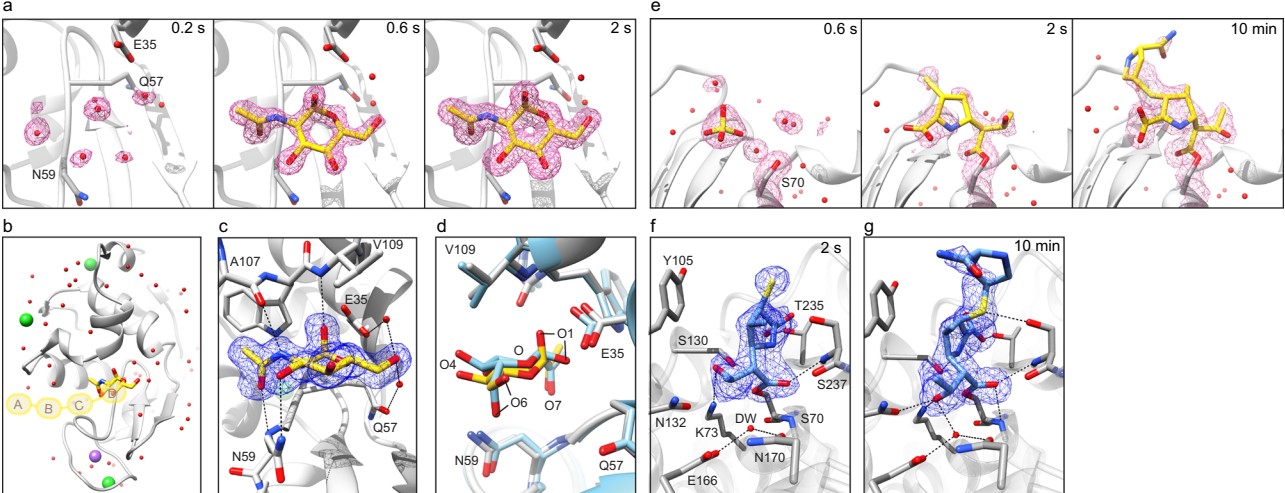

**Fig. 2 Drop-on-drop delivery enables the analysis of N-acetyl-D-glucosamine (GlcNAc) binding to the HEWL active site and catalytic activity of CTX-M-15 by serial X-ray crystallography. a** $2mF_o-DF_c$ electron density maps for HEWL-GlcNAc XFEL structures, displayed at ±1σ contour level and carved at 2 Å around the GlcNAc ligand. The GlcNAc molecule from the 2 s time point structure was used to carve the 200 ms time point map. All maps are at 1.45 Å resolution. **b** Cartoon representation of the HEWL structure (2 s mixing time point, PDB ID 7BHN). GlcNAc is shown as sticks and colored in yellow. Waters, sodium, and chloride ions are shown as red, purple, and green spheres, respectively. Ligand binding subsites A-D (as in PDB ID 5NJR[10]) are indicated by yellow circles. **c**, **d** Close-up active site view of the HEWL structure shown in (**b**). **c** $mF_o-DF_c$ polder OMIT difference density map contoured at ±3σ and carved 1.5 Å around the ligand site. Hydrogen bonding network is highlighted. **d** Superimposition of the 2 s mixing time point structure (yellow and gray) and the single-crystal room-temperature structure of HEWL soaked with GlcNAc obtained by Tanley et al.[25] (light blue, PDB 3TXJ). **e** $2mF_o-DF_c$ electron density maps of CTX-M-15 structure displayed at ±1σ and carved at 2 Å around the ertapenem ligand (pink; 1.55, 1.55, and 1.65 Å for 0.6 s, 2, and 10 min, respectively). The 0.6 and 2 s time points are calculated with XFEL data, whereas the 10 min time point is calculated with room temperature, SSX data collected at Diamond Light Source. **f**, **g** Close-up view at the active site of the structure obtained after 2 s (**f**) and 10 min (**g**) of mixing with ertapenem. Interactions of ertapenem (blue sticks) with residues on the protein main chain (gray sticks) as well as the interactions of the proposed deacylating water (DW, red sphere) with the protein are shown as black dashes. $mF_o-DF_c$ polder OMIT difference density maps are contoured at ±3σ and carved 1.5 Å around the ligand site. Ertapenem has been tentatively modeled and refined into the observed electron density as the (R)-Δ$^1$-pyrroline tautomer for both 0.6 s and 10 min structures (see Supplementary Note 3). Graphic was created using the UCSF Chimera package[54].

synchrotron environment, which will help alleviate restricted access to XFEL facilities that is limited by global capacity. Our method has the potential to increase the popularity of dynamic structural biology studies becoming particularly applicable at next-generation MX beamlines that offer brighter/pink photon beams with sub-millisecond exposure times that will help ensure adequate temporal resolution.

## Methods

**Crystal preparation**. Hen egg white lysozyme (HEWL) crystals were grown by the rapid-mixing batch method[27]. In brief, 2 mL of 50 mg mL$^{-1}$ HEWL (Sigma L4919) dissolved in 20 mM sodium acetate pH 4.6 were pipetted to a 15 mL polypropylene tube and mixed with an equal volume of crystallization buffer (1 M citric acid, 20% (w/v) NaCl, 5% (w/v) PEG 6000, pH 3.0) with vortexing at 22 °C. Crystals nucleated almost immediately and were left for at least 1 h to reach the maximal size (3–5 μm). The data for 0.6 and 2 s datasets were collected from the original crystal slurry (~10$^7$ crystals mL$^{-1}$) whereas the data for the resting and 0.2 s datasets were collected from the same crystal batch, concentrated threefold.

Recombinant CTX-M-15 (in the expression vector pOPINF[28]) was expressed in SoluBL21 (DE3) *E. coli* cells and the His-tagged protein was purified by passage over Ni-NTA resin (Qiagen) in 50 mM HEPES (pH 7.5) and 400 mM NaCl and elution with the addition of 400 mM imidazole[28]. The His-tag was then removed by incubation with 3C protease at 4 °C overnight and capture on Ni-NTA resin. CTX-M-15 protein was loaded onto a Superdex 75 size-exclusion column equilibrated with 50 mM HEPES (pH 7.5) and 150 mM NaCl. Peak fractions were concentrated to 20 mg mL$^{-1}$ by centrifugation. CTX-M-15 crystals were nucleated with seeds generated from crushed ~500 μm sized crystals grown at 20 °C, by sitting drop vapor diffusion in CrysChem 24-well plates[28] (Hampton Research). Drops comprised 1 μL of CTX-M-15 (20 mg mL$^{-1}$, in 50 mM HEPES pH 7.5, 150 mM NaCl) and 1 μL of crystallization solution (0.1 M Tris pH 8.0, 2.4 M (NH$_4$)$_2$SO$_4$) and were equilibrated against 500 μL of crystallization solution. For the seed stock, crystals were crushed, mixed with crystallization solution, and stored at −80 °C until needed. Microcrystals were then grown by sitting drop vapor diffusion in CrysChem 24-well plates at 20 °C. Drops comprised 2 μL of crystallization solution (2.0 M (NH$_4$)$_2$SO$_4$, 0.1 M Tris pH 8.0), 1 μL of seed stock and 2 μL of enzyme

(20 mg mL$^{-1}$) and were equilibrated against 500 μL of crystallization solution. Rod-shaped crystals grew within 24 h, with a maximum width of 5 μm and length of 10–20 μm. They were then harvested and left to settle to increase the crystal concentration to ~8 × 10$^7$ crystals mL$^{-1}$.

**Sample injection and mixing**. Mixing experiments were performed in July 2019 under proposal 2019A8088 at the BL2 instrument at SACLA, Japan[29]. During these experiments, the hutch temperature was typically between 34 and 36 °C. The conveyor belt was used in combination with acoustic droplet ejection (ADE) as described in detail in Fuller et al.[14] with some modifications. The ADE transducer was used to deposit ~3 nL crystal slurry droplets at 30 Hz deposition frequency onto the surface of the Kapton belt (Fig. 1a, Supplementary Fig. 1). The microcrystal suspension was fed into the ADE well from a reservoir through a 250 μm ID fused silica capillary attached to a syringe held in a rotation motor that was programmed to rotate back and forth by 180° to prevent crystal settling. The XFEL master clock running at 30 Hz was used to trigger ADE droplet ejection, which in turn generated a TTL (transistor–transistor logic) signal to trigger a second, piezoelectric injector (PEI) head (PolyPico Technologies, Cork, Ireland) mounted downstream with respect to the belt movement and above the tape. The PEI was used to add an aqueous solution of substrates to the crystal-containing drops in the form of ~120 pL droplets dispensed from a disposable plastic cartridge (100 μm orifice), which was continuously refilled from a syringe pump through a 200 μm ID fused silica capillary. Substrates, N-acetyl-D-glucosamine (GlcNAc, Sigma A8625) and ertapenem (MedChemExpress, USA) were dissolved in water to a final concentration of 50 mg mL$^{-1}$ (226 mM) and 398 mg mL$^{-1}$ (0.8 M), respectively, and filtered through a 0.22 μm syringe filter. Crystal suspensions were filtered through a 100 μm capillary. Both crystal slurry- and substrate-containing syringes were kept at 4 °C to prevent the solutions from degrading.

We tested dispensing substrate at a variety of frequencies, including the maximal achievable frequency of PEI in order to maximize the number of substrate drops hitting the crystal drop and so introducing as much substrate as possible. In order to reduce their consumption, substrates were added in the form of droplet bursts synchronized with the ADE slurry droplets. A camera viewing system was used to manually adjust the PEI trigger delay time that controlled the overlap between the crystal drop and the substrate burst. The calculated theoretical numbers of substrate solution drops merged with each crystal drop for data collected on HEWL and CTX-M-15 are presented in Supplementary Table 4.

Reaction times were varied by changes in Kapton belt velocity, bringing the sample to a fixed beam interaction location. The setup allowed for reaction times between 0.1 and 6 s (600 and 10 mm s$^{-1}$ tape speed, respectively). Our control datasets without substrate additions were recorded using 100 or 300 mm s$^{-1}$ tape speed. The number of substrate drops dispensed per burst (between 10 and 20) slightly exceeded the number required to fully cover the crystal drop. We observed that this greatly simplified the drop synchronization process and resulted in much more stable ejection. Operation in burst mode (10 droplets at 30 Hz) reduced substrate consumption up to 20-fold as compared to continuous delivery (continued dispensing at 6.1 kHz) with 84–258 μL of substrate consumed per dataset. A further reduction in substrate consumption could have been achieved by using a smaller cartridge orifice (for example, 30 μm diameter instead of 100 μm) at the cost of lowering the maximal achievable substrate concentration (50 pL droplets instead of 120 pL) and increasing the probability of clogging if unanticipated particulates are present in the substrate solution.

**Fluorescence measurements of calcium-sensitive dyes.** To experimentally assess substrate mixing and equilibration times through the ADE drops, the drop-on-drop setup was used to record the fluorescence of calcium-sensitive dyes in solution. A dye solution containing 1 mM Fura Red (Ca(II) $K_D$ of 400 nM, AAT Bioquest) and 1 mM Fluo-5N (Ca(II) $K_D$ of 90 μM, Thermo Fisher) was dispensed by ADE at 30 Hz onto the Kapton tape in the form of 4 nL drops. One to five drops of 100 mM CaCl$_2$ solution were dispensed from the PEI onto the ADE drops at 0.5–1 kHz frequency. The PEI was equipped with a 100 μm orifice cartridge, which produced an average drop volume of 60 pL to yield an equilibrium concentration of 1.5–7 mM CaCl$_2$. The fluorescence signal was measured at the same position as the XFEL interaction point with a fiber-coupled spectrometer (Ocean FX). The samples were excited by a 455 nm 10 mW LED (M455F1, Thorlabs). A long-pass filter with a cutoff at 500 nm blocked the scattered excitation light (FEL0500, Thorlabs). The tape speed was varied to adjust the interaction time. The fastest interaction time measured was 200 ms, with measurements constrained by the position of the PEI head and limited by the sensitivity of the optical system. Four scans of 300 ms integration time were averaged per measurement (signal from 10 drops on average). Each measurement was repeated 10 to 50 times with and without addition of CaCl$_2$. As the residence time and thus the fluorescent signal of a single drop in the measured path is inversely proportional to the tape speed, the fluorescence signal was scaled by the inverse tape speed (Supplementary Fig. 5).

To extract the strength of the fluorescence signal of one component, a function containing the single components' normalized spectra scaled by a variable prefactor $s$ was fitted to the averaged fluorescence spectrum from 510 to 750 nm according to Eq. (1) (SciPy package[30], Supplementary Fig. 5):

$$f(\lambda) = s_d * sc_{diode} + s_{fluo} * fl_{fluo} + s_{fura-} * fl_{fura-} + s_{fura+} * fl_{fura+} + c \qquad (1)$$

where $sc_{diode}$ describes the residual diode scatter, $fl_{fluo}$ the fluorescence of the Fluo-5N dye (calcium-dependent amplitude, $\lambda_{max} = 515$ nm), $fl_{fura+}$ and $fl_{fura-}$, the calcium-free ($\lambda_{max} = 670$ nm) and calcium-bound forms ($\lambda_{max} = 640$ nm) of Fura Red, respectively, and $c$—a constant offset. The spectrum of the residual diode scatter was extracted from the highest amplitude measurement without CaCl$_2$ (negligible dye contribution below 560 nm). The spectrum of Fluo-5N was used as given by the provider (Thermo Fisher). The spectra of both Fura Red forms were extracted from fluorescence measurements performed on an 8 μM dye solution in a cuvette either with or without CaCl$_2$ (50 mM). The scaling factors of both calcium-bound dye species ($s_{fluo}$ and $s_{fura+}$) were extracted and plotted versus the interaction time $t$. The time-dependent fluorescence increase was fitted by an exponential function according to Eq. (2):

$$s_{fluo,fura+}(t) = a * (1 - e^{-t/\tau}) \qquad (2)$$

where $a$ is a prefactor and $\tau$ is the rise time. It should be noted that a single exponential rise is insufficient to describe the physical processes of three-dimensional mixing and diffusion, though the fitted rise allows to compare the kinetics of the fluorescent increase with the simulations described below. To find a reasonable fit for the weak fluorescence signal from Fura Red, the results from the two fastest interaction time points were averaged (grayed out points in Fig. 1b).

**Diffusion simulation.** Kinetic simulations of the calcium-fluorescent dye systems were used to characterize the mixing process in the drop-on-drop system. Two extrema are possible: mixing entirely by diffusion and mixing entirely by hydrodynamic flow[31]. The open-access software Kinetiscope (W. D. Hinsberg and F. A. Houle, Kinetiscope, www.hinsberg.net/kinetiscope, 2020) permits both regimes to be examined using a stochastic algorithm for both full three-dimensional reaction–diffusion simulations and for simple mass transfer simulations. The stochastic method is a type of kinetic Monte Carlo algorithm that provides a rigorous solution to the master equation for Markov systems[32,33] and is applicable to reaction–diffusion simulations[34]. In brief, the system is described using particles to represent molecules, located in a system of compartments with specified dimensions. The compartments are coupled by diffusion paths in the 3-D system, and by mass transfer paths when multicompartment systems are used. The reaction scheme comprises discrete reaction and diffusion or mass transfer steps, with corresponding rate constants. The software generates fully spatially resolved

concentration data on an absolute time base, providing insight into how the nano- and mesoscale-level behavior of the system gives rise to experimental observables.

A three-dimensional reaction–diffusion simulation scheme was used to model the fluorescence of the calcium-sensitive dyes in order to predict the behavior of the time-dependent measurements with chemistry solely under diffusive control. This model assumes that the relative velocity of the PEI droplet and its collision with the large ADE drop on the tape make no contribution to the mixing time. The smaller, picoliter-sized PEI droplet(s) rest on top of the larger ADE droplet at time point zero, such that the transfer of species within the combined volume is driven purely by Fickian diffusion, i.e., the diffusion flux is proportional to the negative of the concentration gradient and there is no bulk flow of liquids. Reaction–diffusion simulations were performed to calculate overall reaction rates between calcium and the dyes after contact between ADE and PEI drops. The model was built such that the final volumes and assigned initial concentrations of species in both droplets matched those observed experimentally (4 nL for ADE and 3 × 60 pL for PEI, Supplementary Fig. 2 and Supplementary Table 1). Diffusion coefficients and rate constants were taken from the literature[35–38]; where exact data were unavailable, values were approximated using constants for similar chemical species (Supplementary Table 2). Concentration data as a function of time were averaged over the combined volume of the droplets.

The existing model was then modified to construct an additional three-dimensional reaction–diffusion system using the software Kinetiscope (Supplementary Fig. 2). Species concentrations, reaction kinetics, and total drop volumes were kept constant and matched those used experimentally (Supplementary Tables 1 and 2). The Weber number of the large ADE droplet was first calculated, and three-dimensional idealizations of the internal mixing jets formed within a millisecond after collision described in literature were built using Kinetiscope[39] (Supplementary Table 3). In this altered geometry, the three PEI drops were represented as conical volumes extending into the ADE drop (Supplementary Fig. 4). This was to investigate the effect of partial mixing on the diffusion time of calcium and dyes throughout the combined drop volume. While this system similarly assumes that the movement of species within the drops is entirely diffusive, it allows the simulation to start with the internal cone-like structures formed within milliseconds after impact[39]. The formation of the fluorescent complexes was monitored by extraction of concentration data as a function of time averaged over the total drop volume.

Mass-flow compartmental simulations of free colliding drops were also performed to obtain a lower bound on the mixing time of a process controlled purely by hydrodynamic flow with no diffusive movement of species. The relative velocity of the PEI droplet was estimated from the speed of ejection from the PEI head and the speed of the traveling tape. Calculation of the transfer rate of the calcium ligand was performed for picoliter-sized droplets traveling at this relative velocity and colliding with a 200 μm diameter droplet (Supplementary Table 3)[31]. Inertial mixing was used in multicompartment simulations that include proportional mass flow to evaluate the impact of hydrodynamic flow during the droplet collision on mixing time (Supplementary Fig. 3). In all three cases, the time-dependent concentrations of the fluorescent species (i.e., both calcium-bound dyes) were compared to the experimental fluorescence signals.

**Enzyme assays.** All reactions were carried out in 10 mM HEPES pH 7.5, 150 mM NaCl at 25 °C using Greiner half area 96-well plates and a POLARstar Omega (BMG LabTech) plate reader[28]. Kinetic parameters were calculated and analyzed using GraphPad Prism 6.0.0 for Windows, GraphPad Software, San Diego, CA, USA, www.graphpad.com[28,40]. Steady-state parameters $k_{cat}$ and $K_M$ for ertapenem hydrolysis were calculated by measuring initial rates of ertapenem hydrolysis with 5 μM CTX-M-15 and plotted against ertapenem concentration. Under the same experimental parameters, CTX-M-15 hydrolyses nitrocefin (a β-lactam reporter substrate) with a $k_{cat}$ of ~300 s$^{-1}$ [28]. 5 μM CTX-M-15 was required to initiate the very slow hydrolysis of ertapenem for $k_{cat}$ and $K_M$ determination. $K_{iapp}$ and $k_2/K$ were calculated using direct competition assays[28,40] (under steady-state conditions without preincubation) against the chromogenic reporter substrate nitrocefin[41]. The reciprocals of the initial rates of nitrocefin hydrolysis (at a fixed concentration of 50 μM) by 1 nM enzyme were plotted against ertapenem concentrations ranging from 156 nM to 20 μM and corrected to account for the $K_M$ of nitrocefin[28].

**Diffraction data collection.** Data at SACLA were collected at 34–36 °C (similar to the hutch temperature) using ~450 μJ X-ray pulses of 10 fs length at 10 keV and with X-rays focused to a spot of 1.4 μm FWHM at the sample position. Data were recorded at 30 Hz on a Rayonix MX300-HS detector operating in the 4 by 4 binning mode. Over the course of the SACLA data collection, the average ratio for crystal lattices integrated to images collected ranged from ~6 to 26%. Each dataset was collected within 30–80 min consuming between 162 and 420 μL of microcrystal slurry per one dataset, which corresponds to estimated average sample consumption of 1.40 and 0.27 μg of HEWL and CTX-M-15 per integrable detector frame, respectively.

Room-temperature (21 °C) datasets on reference crystals as well as crystals exposed to substrate for 10 min were collected on beamline I24, Diamond Light Source. Data were recorded on a Pilatus 6 M detector with 10 ms exposure time (5.2 × 10$^{12}$ ph s$^{-1}$) using a beam size of 8 by 8 μm$^2$ and an X-ray energy of 12.4 keV. Sample was mounted in the fixed-target setup[42] with 100 μL of the

microcrystal slurry or 100 µL of 1:1 (v/v) mixture of the slurry and 100 mM ertapenem solution loaded per single chip.

**Data processing**. Lysozyme data reduction was performed on frames identified as hits by Cheetah[43] and achieved using cctbx.xfel[19,44,45] as described previously[19]. HEWL data were indexed with dials.stills_process[46] in space group $P4_32_12$ with the target unit cell of $a = b = 79.3$ Å, $c = 38.2$ Å, $\alpha = \beta = \gamma = 90°$ (subsequently refined as detailed below). Data were merged using cxi.merge[19,44,45] with PDB entry 4ETA[47] serving as the initial reference model. The resolution cut-offs for the final datasets merged with cxi.merge were determined in a standard procedure based on a combination of several criteria, including where the data falls below tenfold multiplicity, where $CC_{1/2}$ no longer decreases monotonically, and where the values of $I/\sigma(I)$ do not uniformly decrease any more[14]. The resolution cut-offs were also confirmed in the 'paired refinement' test done using the PDB_REDO platform[48], which includes the implementation of the original algorithm from Karplus and Diederichs[49]. In the first round of merging, integrated frames from all datasets were pooled together and the unit cell parameters were allowed to refine. This produced a 'composite' MTZ file with the unit cell parameters representing all data ($a = b = 78.8$ Å, $c = 38.0$ Å, $\alpha = \beta = \gamma = 90°$). Next, a new reference model was obtained by running molecular replacement using the same coordinates against the 'combined' MTZ file with Phaser[50]. The new reference model was then used in the second, final merging round in which the target unit cell parameters were fixed and only frames with unit cell parameters lying within 1% of these values were accepted. The reference model used in the final merging step was used as the starting point in structure refinement. The structures were refined via iterative cycles of refinement performed using Phenix[51] and manual model rebuilding using Coot[52]. Insights into ligand binding were obtained by examination of structure-factor amplitude Fourier difference maps, calculated with Phenix by subtracting observed structure-factor amplitudes for the resting dataset from those of the time point dataset, using phases calculated from the resting state model. GlcNAc was placed in the models according to the initial $F_{o\,(time\,point)}−F_{o\,(resting)}$ and $mF_o−DF_c$ maps. Restraints for GlcNAc were generated using the Grade Web Server (http://grade.globalphasing.org, 2020) using the identifier NDG. Data collection details and refinement statistics can be found in Supplementary Table 5.

CTX-M-15 data were indexed and integrated using DIALS[46] (synchrotron data) and cctbx.xfel[19,44,45] (XFEL data). Images were indexed with dials.stills_process in space group $P2_12_12_1$ with the target unit cell $a = 45.3$, $b = 45.9$, $c = 118.5$ Å, $\alpha = \beta = \gamma = 90°$ (subsequently refined as detailed below). Next, datasets were merged with cxi.merge[19,44,45] using PDB entry 6QW8[28] as the initial reference model to provide starting unit cell parameters, which were allowed to refine. Resolution cut-off criteria were assessed in the same way as for the HEWL data. For the second and final round, reference models were obtained in Phaser[50] by running molecular replacement with the same coordinates against the MTZ files from the first round of merging (unit cell parameters were allowed to refine further). The structures were refined via iterative cycles of refinement performed using Phenix[51] and manual model rebuilding using Coot[52] after molecular replacement with Phaser[50]. Geometry restraints for ertapenem were calculated using eLBOW in Phenix[53]. For isomorphous difference density map calculations, a second set of MTZ files merged using fixed unit cell parameters with 1% tolerance was produced, as described above for HEWL. Due to the differences in the unit cell parameters, data from SACLA and DLS I24 had to be treated separately ($a = 44.89$, $b = 45.57$, $c = 117.61$ Å for SACLA; $a = 45.27$, $b = 45.97$, $c = 118.74$ Å for I24). Data collection and refinement statistics details can be found in Supplementary Table 6.

**Reporting summary**. Further information on research design is available in the Nature Research Reporting Summary linked to this article.

## Data availability

Coordinates and structure factors that were generated during the course of this study have been deposited in the Protein Data Bank with the accession codes 7BHK (HEWL SACLA resting state), 7BHL (HEWL SACLA 0.2 s time point), 7BHM (HEWL SACLA 0.6 s time point), 7BHN (HEWL SACLA 2 s time point), 7BH3 (CTX-M-15 SACLA resting state), 7BH4 (CTX-M-15 SACLA 0.6 s time point), 7BH5 (CTX-M-15 SACLA 2 s time point), 7BH6 (CTX-M-15 DLS I24 resting state), and 7BH7 (CTX-M-15 DLS I24 10 min time point). Protein structures used as search models in molecular replacement are accessible in the Protein Data Bank under accession codes 4ETA[47] (HEWL) and 6QW8[30] (CTX-M-15). Source data are provided with this paper.

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

## Acknowledgements

The authors acknowledge the financial support of this work from the National Institutes of Health (NIH) grants GM117126 (to N.K.S.), GM55302 (to V.K.Y.), GM110501 (to J. Y.), GM126289 (to J.K.), and NIH training grant GM133081 (to K.D.S.). Support was provided by the Biotechnology and Biological Sciences Research Council Grant 102593 (to A.M.O.); Wellcome Investigator Award in Science 210734/Z/18/Z (to A.M.O.); Royal Society Wolfson Fellowship RSWF\R2\182017 (to A.M.O.) and by the Director, Office of Science, Office of Basic Energy Sciences (OBES), Division of Chemical Sciences, Geosciences, and Biosciences of the Department of Energy (DOE) (to J.K., J.Y., and V.K.Y.). F.A.H. was supported by the Laboratory Directed Research and Development Program of the Department of Energy's Lawrence Berkeley National Laboratory under DOE OBES under Contract No. DE-AC02-05CH11231. C.L.T. was supported by BBSRC-funded South West Biosciences Doctoral Training Partnership (BB/J014400/1). J.J.A.G.K. was funded by the EPSRC Synthesis 345 for Biology and Medicine CDT(EP/L015838/1) and a Clarendon Scholarship. R.T. was supported by Platform Project for Supporting Drug Discovery and Life Science Research (Basis for Supporting Innovative Drug Discovery and Life Science Research (BINDS)) from AMED under Grant Number JP20am0101070. XFEL data was collected under proposal 2019A8088 at the BL2 instrument at SACLA, Japan. Fixed-target data was collected under proposal mx19458 and mx25260 at Diamond Light Source BLI24.

## Author contributions

P.A., J.F.K., A.M.O., V.K.Y., and J.Y. conceived the experiment which was designed by J.F.K., P.S.S., P.A., and G.L. P.S.S. and I.-S.K. carried out and analyzed fluorescence experiments. R.N.M. and F.A.H. carried out and analyzed the simulations. A.Bu., P.H., and C.L.T. prepared microcrystals. A.Bu., P.S.S., P.A., P.H., G.L., C.L.T., I.B., I.-S.K., A.Bh., A.S.B., N.E.D., J.B., J.J.A.G.K., P.A.L., P.R., S.O., R.T., K.T., J.F.K., and A.M.O. ran the experiment at SACLA. A.Bu., P.A., R.L.O., D.A., J.H.B., B.D., N.E.D., A.E., J.O., S.L.S. S., and T.Z. collected the crystallographic data at I24. A.Bu., P.H., A.Bh., N.E.D., N.K.S., and G.E. analyzed the crystallographic data. A.Bu., P.S.S., P.A., P.H., R.N.M., F.A. H., J.S., C.J.S., J.F.K., and A.M.O. analyzed all the data and wrote the initial manuscript. All authors provided critical feedback and contributed to the final version.

## Competing interests

G.L. is the co-founder of PolyPico Technologies Ltd. All other authors declare no competing interests.
