## [Peer Review File · Nature Communications]

REVIEWER COMMENTS

Reviewer #1 (Remarks to the Author):

The object of any time-resolved crystallography experiment is generally the identification of kinetic mechanisms that evolve in a crystal and the characterization and evolution of the population of states that comprise those mechanisms. Several groups have begun to develop technologies that try to explore the boundaries of what is measurable at both XFEL and synchrotron sources – the development of these new methods can be both complicated and challenging. The major goal of this study is to create a flexible system for serial crystallography that will ensure efficient data collection without the need for optical excitation while ensuring a wide gamut of timescales can be measured. The manuscript presents a modification to the “drop-on-tape” design with the addition of another piezoelectric droplet ejection head which allows for the dispensation of multiple picolitre droplets on a larger nanoliter droplet containing a slurry of crystals – allowing for a “drop-on-drop” mixing approach. Overall, the paper is clear and well written, the concept is sound, the data are convincing, and this is another novel piece of technology with the potential to become useful in the long-term as the field of time-resolved crystallography grows. Therefore, assuming my comments below can be answered, I would recommend publication in Nature Communications.

There is an apparent appeal of XFEL sources for both serial and time-resolved approaches, given the ultrashort pulse lengths and peak brightness, it's conceivable why beamtime on these sources is in high demand. Since, the measurable time-resolution should dictate the choice of radiation source. I would like the authors to comment whether their system can be used at synchrotrons. Are there any drawbacks, given the timescales that are capable with their setup this should be completely possible? Are there limitations to this setup at synchrotrons? Given the wider availability and use of microfocus beamlines, their setup can potentially become available to a larger array of users. I am missing this discussion.

Given the large size of the nanoliter sized droplet containing the crystal slurry, minimizing this volume ensures lower background scattering and will speed up diffusion. Given that the crystals are not always homogeneously distributed throughout the droplet, what would the effects on the diffusion time be if the crystals were clumped together or isolated in one particular portion of the droplet furthest away from where the picolitre droplets make contact?

The nanoliter droplets contain a slurry of crystals. Multiple diffraction patterns on a single image therefore become inevitable while imaging a single droplet. If so, how dense is the slurry and how were these dealt with?

Examining the data statistics, with CTX-M-15 Resting, roughly ~4500 lattices were merged, can the authors comment on why the R-factors are considerably higher than the other structures, which were exposed to their ligand?

What is the authors cutoff criteria for their high-resolution CC1/2 values, as some structures in the highest resolution shell have very low cc1/2 values, a few below 5%, and one less than 1%. Did paired-refinement at these cutoffs show an improvement in the refinement behaviour?

What was the crystal size distribution, how was this controlled? If there was a large discrepancy in crystal size 1-3x volume changes or if multiple crystals overlaid on top of one another in a droplet would result in a larger total crystal volume with correspondingly increased diffusion times, do the authors account for this?

I review non-anonymously, Pedram Mehrabi (MPSD)

Reviewer #2 (Remarks to the Author):

The authors present a nice time-resolved serial crystallography study using a drop-on-demand delivery system. They claim to be able to study chemical reaction in protein crystals using diffusing mixing in nanoliter droplets of a crystal slurry. This is a very interesting study which focuses on two proteins, the well known lysozyme and CTX-M15 kinetics. Although there are several different methods published based on drop-on-demand this is the first of its kind (to my knowledge) to demonstrate its ability to show chemical mixing and studying the kinetics of enzymes, which is of great interest to biologist. Current time resolved methods are limited to pump-probe experiments which limits the variety of proteins we can study. The authors claim this method is more feasible in terms of sample consumptions by using microdroplets. This is a promising technique which does drastically reduced the volume of sample to collect complete data sets.

Major comments:

1. The authors provide a nice representation of the time series data of the lysozyme enzymatic reaction with its ligand from rest state up to 2s. The structural analysis clearly shows the gradual docking of the ligand into the active site, with the high-resolution structural information to support this. They also demonstrate the method works for CTX-M-15 however the complete reaction of ligand docking was not detected, as the data collection time did not exceed the 2s mark for the XFEL experiment and the authors seem to have missed the completed reaction time. Instead they used a static time point collected on a fixed target at a different lights source which is OK, but it would have been nice to see the complete reaction at SACLA using the drop-on-demand source. Was there a particular reason for this? Is the drop and demand method limited to a certain duration of kinetics or are we able to capture longer time frames than 2s. This was not clear in the manuscript. It would also help to state the current limitations of this device if any? What are the maximum droplets sizes possible before the diffusion reactions rates are affected? Despite this, their results do capture a different state of the ligand at 2s. The question however still remains, does the reaction go to the completion of what was seen in the fixed target experiment and are there more intermediate states.

2. What would make this article more informative is the specific Kinetic values for the enzyme and ligands presented in the supplementary information to be inserted within the main text (p6, line 132). What possible reasons could be associated with not seeing the completed reaction within the 2s time frame of XFEL data collection, given that the enzymatic data shows this should have been possible?

3. The mixing simulations presented in the paper show very different results to the experimental data. From my reading of the manuscript, two types of mixing have been simulated, diffusion and hydrodynamic. Fig 1b shows both simulations along with experimental fluorescent mixing studies. The hydrodynamic simulation looks to be too rapid and the diffusion looks to be too slow, compared with the experimental fluorescents data. Given the fact that the CTX-M-15 reaction was not completed within 2s (even in excess ligand concentrations) what conclusion can be drawn from the simulations?

Minor comments:

P3, line 54: '...using the appropriately small crystals,....' Please specify what you consider this range to be. I think this is an important number which needs to be considered for this type of experiment. Are we talking about nm or sub-micron?

P6, line 132: 'ligand affinity (uM vs, mM)...'

It would be helpful to have the actual #s here to make an easier comparison.

P8, line 195, Were the substrate solutions filtered prior to mixing?

P9, line227: It is stated in the manuscript that ' volume of 60pl to yield equilibrium...'

It is not clear in the manuscript why 60pl drop size was used for the CaCl₂ solution and 120pl was used for the HEWL experiment. Also, given that the same size 100um orifice was used how was the volume of the drop altered.

P10, line 23, '.....normalized spectra) scaled'

Remove the bracket.

Figure 1: It may be helpful to label the PEI and ADE on the image.

Reviewer #3 (Remarks to the Author):

In the history of biochemistry, many researchers have studied enzymes, but it has been difficult to completely elucidate their chemical reactions due to the limitations of the technology to visualize the movement of molecules during their catalyses.

This situation is now changing with the development of XFEL technology in the last decade.

Time-resolved serial femtosecond crystallography (tr-SFX) is the only technique that can visualize the movement of biological macromolecules during function with atomic/femtosecond-time resolutions under damage-free room temperature states.

However, in tr-SFX using conventional injectors, it was troublesome to optimize the measurement conditions by repeated trial and error for each type of protein. Furthermore, the consumption of a huge amount of sample was a problem.

In the present work, Butryn et al. have succeeded in overcoming the problems by demonstrating the application of the drop-on-drop method developed from the drop-on-tape XFEL sample delivery system to enzyme-catalyzed reactions in microcrystals.

I appreciate the efficiency of this measurement system and its potential versatility for a wide variety of enzymes.

I request that the authors answer the following questions/comments.

1. The microcrystals of HEWL and CTX-M-15 were both obtained with salt-based precipitants, and their slurry seems to be low viscosity. I expect the authors present reference materials in the present or the future paper that evaluate the diffusion efficiency when using various highly viscous slurries (e.g. 20% PEG4000) in contrast to the drop size.

2. If possible, the authors should prepare a movie showing the efficient mixing of PEI drops on a ADE drop as supplementary data, for a detailed understanding of the measurement system. It does not have to be a movie of the actual measurement at SACLA, but it can be a movie of fluorescent dye taken offline.

3. In Supplementary Figure 3, it is clear that the contrast between blue and red in the 700 ms figure of (a) is different from that in the other figures in (a). What is the cause of this?

4. I ask a question to gain a better understanding of the quality of the structural data in this study. In Supplementary Table 2, the refinement of the four types of HEWL drop-on-drop data is done at about 1.45Å resolutions, and the CC1/2 values of the outer shell are only 5.4-0.3%. What is the resolution when the CC1/2 values of the outer shell is 50%? Or the authors should show a Table of CC1/2 for each resolution shell.

5. Similar question as 4. In Supplementary Table 3, The CTX-M-15 drop-on-drop data are refined at a

resolution of about 1.6 Å. The CC1/2 values of the outer shell are 10.3-2.2%. What is the resolution of the outer shell when its CC1/2 values are 50%? I would like to see a table of CC1/2 values for each resolution shell.

6. In supplementary Table 4, the k_{cat} value seems to be quite low. What is the k_{cat} in the solution state, not in microcrystals? What is the optimal pH of the enzyme?

NCOMMS-21-04895-T. An on-demand, drop-on-drop method for studying enzyme catalysis by serial crystallography.

We thank all the reviewers for their very positive comments. The comments are copied below (in black) and our responses are in blue. We have addressed all the comments fully, and we think the suggested changes and points for discussion have substantially improved the manuscript.

Reviewer #1

The object of any time-resolved crystallography experiment is generally the identification of kinetic mechanisms that evolve in a crystal and the characterization and evolution of the population of states that comprise those mechanisms. Several groups have begun to develop technologies that try to explore the boundaries of what is measurable at both XFEL and synchrotron sources – the development of these new methods can be both complicated and challenging. The major goal of this study is to create a flexible system for serial crystallography that will ensure efficient data collection without the need for optical excitation while ensuring a wide gamut of timescales can be measured. The manuscript presents a modification to the “drop-on-tape” design with the addition of another piezoelectric droplet ejection head which allows for the dispensation of multiple picolitre droplets on a larger nanoliter droplet containing a slurry of crystals – allowing for a “drop-on-drop” mixing approach. Overall, the paper is clear and well written, the concept is sound, the data are convincing, and this is another novel piece of technology with the potential to become useful in the long-term as the field of time-resolved crystallography grows. Therefore, assuming my comments below can be answered, I would recommend publication in Nature Communications.

There is an apparent appeal of XFEL sources for both serial and time-resolved approaches, given the ultrashort pulse lengths and peak brightness, it's conceivable why beamtime on these sources is in high demand. Since, the measurable time-resolution should dictate the choice of radiation source. I would like the authors to comment whether their system can be used at synchrotrons. Are there any drawbacks, given the timescales that are capable with their setup this should be completely possible? Are there limitations to this setup at synchrotrons? Given the wider availability and use of microfocus beamlines, their setup can potentially become available to a larger array of users. I am missing this discussion.

We totally agree with the reviewer. In the first version of the manuscript, we focused entirely on tr-SFX and the discussion on applicability of the drop-on-drop method at synchrotron sources and the concept of tr-SSX was not included. In the revised version, we made changes throughout the manuscript, where we emphasize that the drop-on-drop, as well as time-resolved crystallography experiments in general, are as applicable to synchrotron sources as they are to XFELs (especially in the Introduction and Discussion sections, for example: P3, line 72; P3/4, lines 75-78; P4, line 85; P7, lines 167-173).

The main bottleneck in accommodating methods like drop-on-drop at synchrotron sources is the minimum exposure time of the crystals to X-rays that is required at MX beamlines in order to obtain enough signal, which automatically limits the time resolution of the method.

Longer exposure times also come with the additional problem of radiation induced damage to the crystals at room temperature. With new upgrades at MX beamlines these problems can be mitigated, when sub-millisecond exposure times will become routine (P7, lines 167-173). Moreover, presently it is difficult to accommodate the setup (designed for XFEL beamlines) on most MX endstations at synchrotrons because of its size. With this in mind and also to make it easier to transport, a smaller version of the setup is being designed and will be available soon.

Given the large size of the nanoliter sized droplet containing the crystal slurry, minimizing this volume ensures lower background scattering and will speed up diffusion. Given that the crystals are not always homogeneously distributed throughout the droplet, what would the effects on the diffusion time be if the crystals were clumped together or isolated in one particular portion of the droplet furthest away from where the picolitre droplets make contact?

These are all valid general concerns and will have to be dealt with on a case-by-case basis.

Diffusion time is proportional to the square of diffusion distance. If the particle movement is caused only by diffusional flux, that creates a gradient of the substrate through the drop. For a droplet size in the few nL range, the time to reach equilibrium would be then significantly longer than the average time of the enzymatic reaction. As our simulations show, however, the impact of droplet collision plays a major role in speeding up the rate at which substrate is distributed through the drop. Consequently, we think that the distribution of crystals in the drop is less of a concern in that case as the equilibration of the ADE droplet with the ligand is faster than the delay times we probed in our study.

We do not have a way to account for increased diffusion times caused by crystal clustering or the presence of crystals that are much bigger than the average, except to avoid these issues to begin with. We noticed that crystals tend to cluster with prolonged storage and therefore whenever possible we grow the crystals on site just before the measurement time. Also, crystal samples are filtered in order to remove any big crystals/larger crystal clusters and the whole syringe used for sample delivery is shaken constantly. We note that this is not specific to the drop-on-drop system and any sample delivery system for tr-SFX/SSX faces similar issues.

The nanoliter droplets contain a slurry of crystals. Multiple diffraction patterns on a single image therefore become inevitable while imaging a single droplet. If so, how dense is the slurry and how were these dealt with?

Crystal slurries that we used in this study had between 10^7 to 10^8 crystals/mL. This concentration, assuming uniform crystal distribution, would result in 30-300 crystals/3 nL ADE drop. Multiple diffraction patterns are therefore, as pointed out by the reviewer, potentially unavoidable. We optimized the crystal concentration to achieve the maximum possible indexing rate, while at the same time ensuring that the number of multiple lattices on each detector frame is within the range that can be handled by the indexing software.

This is very well reflected in the ratio between integrated patterns and frames with indexable patterns. For all datasets analysed in this study, the average integration rate was above 100%. For example, the maximal integration rate was as high as 226% for one of the CTM-M-15 datasets from SACLA (i.e., on average there were 2.26 diffraction patterns integrated from one frame that was indexable with DIALS).

Not every software package will work well when multiple equally strong diffraction patterns are present on the same image (especially for samples with large unit cells, e.g. 200 Å and larger), but the DIALS framework is able to address this challenge well (Acta Crystallogr D Biol Crystallogr. 2014 Oct;70(Pt 10):2652-66). In brief, after the spot-finding routine, a crystal setting matrix compatible with the unit cell that indexes as many observed centroids as possible is first analysed and refined. Once refinement has converged, any remaining unindexed reflections may be analysed for further lattices. In subsequent iterations, joint refinement of the crystal lattices is performed. This process may be repeated until either an insignificant number of unindexed reflections remain, or no further lattices can be identified. If at any stage refinement does not converge, the most recently identified lattice is discarded and only those lattices which were refined successfully are reported.

Examining the data statistics, with CTX-M-15 Resting, roughly ~4500 lattices were merged, can the authors comment on why the R-factors are considerably higher than the other structures, which were exposed to their ligand?

As pointed out by the reviewer, the CTX-M-15 resting state dataset from SACLA contains less than one third of the merged lattices than the other CTM-X-15 datasets included in this manuscript (4,502 vs 15,151 - 18,661 patterns). This results in a relatively low overall multiplicity (26.45), low total $CC_{1/2}$ (83.3%) and high total R_{split} (44.10%) values. Merging a similar, limited subset of patterns from other, bigger datasets results in similarly poor merging statistics. We believe that the relatively high R_{work}/R_{free} is a result of the low number of indexed patterns comprising this dataset. We estimated that for this particular sample, as many as 10,000-15,000 lattices are required in order to obtain a dataset of similar quality as was obtained for the ligand exposed conditions. Unfortunately, due to extreme time constraints during our XFEL experiment at SACLA, it was not possible to collect as many images on the unperturbed CTM-M-15 crystals as we would have liked to. Although we did collect another SSX resting state structure at the Diamond Light Source, the incompatibility of the unit cell parameters did not allow for using this dataset as the reference for isomorphous difference density maps. Also, we think that using a reference dataset collected on exactly the same batch of crystals, under the same experimental conditions is a good practice.

What is the authors cutoff criteria for their high-resolution $CC_{1/2}$ values, as some structures in the highest resolution shell have very low $cc_{1/2}$ values, a few below 5%, and one less than 1%. Did paired-refinement at these cutoffs show an improvement in the refinement behaviour?

The data was merged using the program *cxi.merge* from the *cctbx.xfel* package. The resolution cut-offs for the final datasets merged with *cxi.merge* are determined in a

standard procedure based on a combination of several criteria, including where the data falls below ten-fold multiplicity, where $CC_{1/2}$ no longer decreases monotonically and where the values of $I/\sigma(I)$ do not uniformly decrease any more (Nat Methods. 2017 Apr; 14(4): 443–449). To enable easier inspection of merging results, we included output merging statistics from *cxi.merge* for each resolution shell (Tables 1-9 below). If the reviewer and/or editor consider it appropriate, we can incorporate the nine tables into the supplemental information. The low $CC_{1/2}$ in the highest resolution shells is a consequence of how image data is integrated in the processing pipeline (per-image $I/\sigma(I)$ cutoff), and therefore, cannot be directly compared with numbers obtained using other processing software, e.g. CrystFEL. A detailed explanation of the differences between the approach by the two software packages can be found in the SI material of Ibrahim et al, Proc Natl Acad Sci U S A. 2020 Jun 9;117(23):12624-12635.

Validity of cutoff selection criteria, as the reviewer suggested, can be confirmed by ‘paired refinement’. Similar to our other studies (Proc Natl Acad Sci U S A. 2020 Jun 9;117(23):12624-12635 or Proc Natl Acad Sci U S A. 2020 Jan 7; 117(1): 300–307), we cross-checked our applied resolution cutoffs by applying a ‘paired refinement’ procedure to see whether including data from the highest resolution shells has a positive effect on the refinement as compared to more conservative cutoffs. This test was done using the PDB_REDO platform, which includes the implementation of the original ‘paired refinement’ algorithm from Karplus and Diederichs (Science. 2012 May 25;336(6084):1030-3, IUCrJ. 2014 May 30;1(Pt 4):213-20). The ‘paired refinement’ test suggested the same resolution cutoffs for four out of nine datasets included in this publication. For the remaining five, the suggested resolution cutoffs were 0.04-0.1 Å lower. Taking into account that there is always a bit of variability because of the way binning is done and that PDB_REDO utilizes Refmac for refinement while our original refinements were performed with Phenix, we conclude that the results of ‘paired refinement’ support selection of our resolution cutoffs. Interestingly, for the SACLA HEWL 0.6 s dataset (which has the lowest $CC_{1/2}$ value in the highest resolution shell of 0.3%) ‘paired refinement’ gives the same resolution cutoff as the current cutoff. Conversely, for the CTX-M-15 resting state structure from I24 ($CC_{1/2}$ value of 56.9% in the highest resolution shell), ‘paired refinement’ suggests slightly lower resolution cutoff (1.71 Å instead of 1.65 Å). This is not intuitive and therefore, the assumption that low $CC_{1/2}$ in the high resolution shells equates to low quality data and that high $CC_{1/2}$ automatically means higher data quality are not always valid. The benefits of merging protocols applied by cctbx protocols can be demonstrated by electron density map analysis (Proc Natl Acad Sci U S A. 2020 Jun 9;117(23):12624-12635), where it becomes clear that including “weak” data has no perceptible negative influence on the map quality and can in fact help pick out important subtle features in the maps.

We updated the “Data processing” section in “Online methods” by adding the additional information about the applied resolution cutoff criteria and ‘paired refinement’ (P15, lines 367-372; P16, lines 392-393).

What was the crystal size distribution, how was this controlled? If there was a large discrepancy in crystal size 1-3x volume changes or if multiple crystals overlaid on top of one another in a droplet would result in a larger total crystal volume with correspondingly increased diffusion times, do the authors account for this?

Our batch crystallization protocols for lysozyme produce slurries that are characterised by a very uniform crystal size distribution along all three edges. This is not the case for CTX-M-15, or any other 'real life' sample that we have worked with. Most of the microcrystals that we grow produce elongated forms of crystals. This is also the case for CTX-M-15 that produces rods/needles with an average edge size of 15 μm . We try to ensure as uniform crystal size distribution as possible by using seeding protocols, which seems to work well for our samples. Such a crystal shape is not particularly ideal but at least it allows for fast substrate diffusion from two directions. As mentioned earlier, we minimize crystal clustering effects by using freshly prepared crystal slurries that are filtered before loading into syringes. It should also be noted that the spread observed in crystal size is expected to have a small effect on diffusion times compared to the delay times examined in this study.

Reviewer #2

The authors present a nice time-resolved serial crystallography study using a drop-on-demand delivery system. They claim to be able to study chemical reaction in protein crystals using diffusing mixing in nanoliter droplets of a crystal slurry. This is a very interesting study which focuses on two proteins, the well know lysozyme and CTX-M15 kinetics. Although there are several different methods published based on drop-on-demand this is the first of its kind (to my knowledge) to demonstrate its ability to show chemical mixing and studying the kinetics of enzymes, which is of great interest to biologist. Current time resolved methods are limited to pump-probe experiments which limits the variety of proteins we can study. The authors claim this method is more feasible in terms of sample consumptions by using microdroplets. This is a promising technique which does drastically reduced the volume of sample to collect complete data sets.

Major comments:

1. The authors provide a nice representation of the time series data of the lysozyme enzymatic reaction with its ligand from rest state up to 2s. The structural analysis clearly shows the gradual docking of the ligand into the active site, with the high-resolution structural information to support this. They also demonstrate the method works for CTX-M-15 however the complete reaction of ligand docking was not detected, as the data collection time did not exceed the 2s mark for the XFEL experiment and the authors seem to have missed the completed reaction time. Instead they used a static time point collected on a fixed target at a different lights source which is OK, but it would have been nice to see the complete reaction at SACLA using the drop-on-demand source. Was there a particular reason for this? Is the drop and demand method limited to a certain duration of kinetics or are we able to capture longer time frames than 2s. This was not clear in the manuscript. It would also help to state the current limitations of this device if any? What are the maximum droplets sizes possible before the diffusion reactions rates are affected? Despite this, their results do capture a different state of the ligand at 2s. The question however still remains, does the reaction go to the completion of what was seen in the fixed target experiment and are there more intermediate states.

2. What would make this article more informative is the specific Kinetic values for the enzyme and ligands presented in the supplementary information to be inserted within the main text (p6, line 132). What possible reasons could be associated with not seeing the completed reaction within the 2s time frame of XFEL data collection, given that the enzymatic data shows this should have been possible?

Answer to points 1 and 2:

The longest mixing time that we could have achieved during the SACLA XFEL experiment was in principle 6 s (stated on P4, line 95; P9, line 229). This is determined by the slowest tape speed and the position of the PEI head. Several things need to be considered before attempting to probe longer timepoints. Please note that our setup, in the form we used at SACLA, was not enclosed and the humidity was not controlled. Moreover, temperature in the experimental hutch was typically between 36-38 °C (this needs to be better controlled by the facilities). This was not only causing rapid droplet evaporation (which inevitably was leading to crystal damage) but also increased salt crystal formation from the mother liquor containing 2 M ammonium sulphate. In order to mitigate the problems resulting from rapid evaporation, crystal slurries can be supplemented with glycerol solution which slows down evaporation. Adding glycerol comes however at a price of increased viscosity and therefore is counterproductive for usage in diffusion-driven experiments. We found that the 2 s timepoint was the longest time we could reliably use under these particular experimental conditions. The next generation of the drop-on-drop setup will be equipped with humidity control and, given that the position of the PEI head can be moved upstream, will allow us to achieve mixing times of up to ~10 s. We included a statement in the conclusion part of the manuscript to describe the possible range of crystal sizes and delay times that can be targeted with the system (P7, lines 159-166).

In CTX-M-15 data we observed an empty active site after 0.6 s and already formed acyl-enzyme complex with partial (74%) occupancy after 2 s. This is despite significant excess of the substrate over protein in the solution, which suggests that the first step of the reaction, i.e. acyl-enzyme formation, happens on a very fast time scale that is beyond the time resolution supported by the version of the drop-on-drop system presented in this manuscript. Moreover, the 2 s acyl-enzyme structure is essentially the same as the 'steady state' soaked structure from the fixed target. Data collected on 'steady state' soaked samples collected with longer incubation times (between 15 min and 24 h, not shown here) show that the acyl-enzyme structure remains largely unchanged. The only difference is that, at higher resolution, more than one tautomer can be distinguished, but the presence of a particular tautomer is not related to incubation time. Finally, the product actually never leaves the active site. The only consistent difference that we notice between the 2 s, 10 min and longer incubation time structures is the increasing ligand occupancy. Therefore, we would expect a > 2s structure to be largely similar to the 2 s dataset with slightly increased ligand occupancy.

CTX-M-15 behaviour is in contrast to the lysozyme data that shows > 60% ligand occupancy after 0.6 s, despite very low ligand affinity. This suggests that there is something particular about the CTX-M-15-ertapenem system that prevents the ligand from entering and leaving

the active site. We think that this particular characteristic must be a combination of protein, crystal lattice and crystallization buffer properties and is not related to the physical process of diffusion in solution itself. In other words, although the CTX-M-15 data helped us to demonstrate capabilities of the drop-on-drop method itself, we admit that this crystal system might not be ideal for studying with drop-on-drop methods. It is impossible to tell what is the main cause responsible for this behaviour, especially as reliable methods for characterizing reaction rates in slurries of microcrystals are essentially non-existent.

As requested, we included binding constants directly in the main text of the revised manuscript. Now the text reads: “ (...) 11.6 μ M for CTX-M-15-ertapenem [Supplementary Table 8] vs. 47.6 mM for HEWL-GlcNAc²³ (...)” (P6, lines 142-143).

3. The mixing simulations presented in the paper show very different results to the experimental data. From my reading of the manuscript, two types of mixing have been simulated, diffusion and hydrodynamic. Fig 1b shows both simulations along with experimental fluorescent mixing studies. The hydrodynamic simulation looks to be too rapid and the diffusion looks to be too slow, compared with the experimental fluorescent data. Given the fact that the CTX-M-15 reaction was not completed within 2s (even in excess ligand concentrations) what conclusion can be drawn from the simulations?

In this study, we used numerical simulations as a way to help us understand and interpret our experimental fluorescence data, which showed that the equilibration process after drop collision is faster than what would be predicted from pure diffusion, i.e. much faster than we expected. It was very important for us to find additional means to confirm and help describe this effect, as it was basically the prerequisite for this method to serve its purpose. We agree with the reviewer that our attempts at describing what is happening in the system after droplets collide do not provide a complete model for this process. We express this opinion in Supplementary Discussion, where we state that future work is clearly required to provide a better description of this process. Our conclusion is that the mixing in drops can be described as a combination of diffusion and hydrodynamic mixing (internal jets) and this is a good start for any future studies on droplet-based methods. We have for example started working on improvements to our fluorescence-based detection system to allow us to probe smaller droplet volumes on faster mixing time scales, which will better match timescales typically explored in simulations.

Please note that the fluorescence measurements and simulations were performed on fluorescent dye and calcium chloride solutions. Therefore, the results obtained on this simplified experimental system cannot be directly transferred to other systems. We do not think that the timescales obtained represent well what is happening in the drop composed of microcrystal slurry and we also do not claim that in the manuscript. Since any simulation uses experimentally determined diffusion coefficients, in the absence of these experimental values for ertapenem in 2 M ammonium sulphate solution, we are not able to provide an approximation for this system. Our assumption would be that the mixing time for ertapenem is slower than that of calcium chloride. Even if we were able to provide an approximation, it would be far from experimental observations. Simulations do not account for diffusion through the crystal lattice which, as we explained in our replies to questions 1

and 2, seems to play a major role in defining overall diffusion rates and will be highly protein-substrate specific.

Minor comments:

P3, line 54: '...using the appropriately small crystals,....' Please specify what you consider this range to be. I think this is an important number which needs to be considered for this type of experiment. Are we talking about nm or sub-micron?

We added missing detailed information to the text (now P3, line 68). The text reads "(...) using appropriately small crystals (< 5 μm), millisecond diffusion times are achievable (...)". This is based on the estimations described in M. Schmidt, *Advances in Condensed Matter Physics* 2013, 167276, doi:10.1155/2013/167276 where 1 ms diffusion time was calculated for glucose diffusing into a $3 \times 4 \times 5 \text{ m}^3$ crystal.

P6, line 132: 'ligand affinity (μM vs, mM)...'

It would be helpful to have the actual #s here to make an easier comparison.

We included this information directly in the main text. Now the text reads: "(...) 11.6 μM for CTX-M-15-ertapenem [Supplementary Table 8] vs. 47.6 mM for HEWL-GlcNAc²³ (...)" (P6, lines 142-143).

P8, line 195, Were the substrate solutions filtered prior to mixing?

That is correct. We added this missing information to the manuscript. The text reads now: "(...) were dissolved in water to a final concentration of 50 mg/mL (226 mM) and 398 mg/mL (0.8 M), respectively, and filtered through a 0.22 μm syringe filter. Crystal suspensions were filtered through a 100 μm capillary (...)" (P9, lines 217-219).

P9, line 227: It is stated in the manuscript that 'volume of 60pl to yield equilibrium...'

It is not clear in the manuscript why 60pl drop size was used for the CaCl_2 solution and 120pl was used for the HEWL experiment. Also, given that the same size 100um orifice was used how was the volume of the drop altered.

The size of the droplets produced by any cartridge will vary depending on the factors like orifice size, dispensing frequency, temperature, surface tension, viscosity and concentration of the substance. For any given cartridge this range will be very broad. In our experience, dispensing from a 100 μm orifice cartridge can produce droplets in the range of 30 to 200 μL and it will also vary slightly from cartridge to cartridge. The volume of the droplets is calculated directly by the software controlling the piezoelectric ejector based on image analysis of the generated droplets. 0.1 M calcium chloride solution, which was used as substrate in fluorescence experiments, manifested water-like behaviour and dispensed as $\sim 60 \text{ pL}$ droplets. To allow direct comparison, we used the same 60 μL droplet size in numerical simulations. The dispensing behaviour of GlcNAc and ertapenem observed during this particular experiment at SACL A was very different to calcium chloride. These substrates

were dissolved close to their maximal solubility limit and were much more viscous than calcium chloride solution. We also think that the very high temperature in the experimental hutch (36-38 °C) had a significant impact on the dispensing behaviour. As a direct calculation from the image processing software during the experiment was not possible due to very poor lighting conditions, we estimated the volume of GlcNAc and ertapenem droplets to be around 120 pL (based on consumption data and the number of generated droplets) and we used this value for all calculations as listed in the revised Supplementary Table 4.

P10, line 23, '.....normalized spectra) scaled'
Remove the bracket.

The bracket was removed.

Figure 1: It may be helpful to label the PEI and ADE on the image.

We modified the labelling on Figure 1 to allow easier identification of the components.

Reviewer #3

In the history of biochemistry, many researchers have studied enzymes, but it has been difficult to completely elucidate their chemical reactions due to the limitations of the technology to visualize the movement of molecules during their catalyses. This situation is now changing with the development of XFEL technology in the last decade. Time-resolved serial femtosecond crystallography (tr-SFX) is the only technique that can visualize the movement of biological macromolecules during function with atomic/femtosecond-time resolutions under damage-free room temperature states. However, in tr-SFX using conventional injectors, it was troublesome to optimize the measurement conditions by repeated trial and error for each type of protein. Furthermore, the consumption of a huge amount of sample was a problem.

In the present work, Butryn et al. have succeeded in overcoming the problems by demonstrating the application of the drop-on-drop method developed from the drop-on-tape XFEL sample delivery system to enzyme-catalyzed reactions in microcrystals. I appreciate the efficiency of this measurement system and its potential versatility for a wide variety of enzymes.

I request that the authors answer the following questions/comments.

1. The microcrystals of HEWL and CTX-M-15 were both obtained with salt-based precipitants, and their slurry seems to be low viscosity. I expect the authors present reference materials in the present or the future paper that evaluate the diffusion efficiency when using various highly viscous slurries (e.g. 20% PEG4000) in contrast to the drop size.

We strongly agree with the reviewer that it will be extremely beneficial to evaluate the diffusion efficiency depending on various factors, including ADE droplet size, mother liquor composition, or PEI droplet number, size, or velocity. This will allow us to select the most optimal experimental parameters as well as a crystal sample system that is the most

promising. The fluorescence-based method that we demonstrated in this study has a great potential for enabling an extensive characterisation of all the above mentioned parameters in the context of the drop-on-drop technique. In addition, we will run numerical simulations that shed light on how different parameters affect diffusion and mixing in colliding drops. For this, empirically-determined diffusion coefficients in various crystallization media and fast imaging of the drop collision event will need to be determined. Our efforts to characterise the drop-on-drop method in more depth have been unfortunately heavily negatively affected by the SARS-CoV-2 pandemic outbreak as our access to the labs has been largely restricted or entirely blocked. We believe that such an in-depth characterization of the drop-on-drop method can easily serve us as a topic for a whole separate manuscript.

2. If possible, the authors should prepare a movie showing the efficient mixing of PEI drops on a ADE drop as supplementary data, for a detailed understanding of the measurement system. It does not have to be a movie of the actual measurement at SACLA, but it can be a movie of fluorescent dye taken offline.

As suggested by the reviewer, we added to the supplementary material two high-speed videos showing a typical example of ADE and PEI droplets merging in a drop-on-drop experiment (Supplementary Video 1 and 2). In Supplementary Video 1 we show a 3 nL ADE droplet merging with a burst of ten PEI droplets dispensed at 1 kHz. In Supplementary Video 2 a similar experiment was recorded, but the camera was synchronised with droplet ejection to visualise positional and temporal precision of PEI and ADE drop dispensing.

3. In Supplementary Figure 3, it is clear that the contrast between blue and red in the 700 ms figure of (a) is different from that in the other figures in (a). What is the cause of this?

We thank the reviewer for pointing out this inconsistency. We carefully re-analysed the data used for generating Supplementary Figure 3. We found out that an error had crept in when we were calculating normalized B-factors for the 600 ms (previously incorrectly labelled as 700 ms) timepoint lysozyme structure. This error caused all B-factors in that structure to be undervalued. This has been corrected in the revised version of Supplementary Figure 3 (now Supplementary Figure 8), where all panels with displayed structures were replaced. We also updated the scale bar and averaged normalized B-factor of selected residues (now in Supplementary Table 7, P19 in the Supplementary Information).

4. I ask a question to gain a better understanding of the quality of the structural data in this study. In Supplementary Table 2, the refinement of the four types of HEWL drop-on-drop data is done at about 1.45Å resolutions, and the CC1/2 values of the outer shell are only 5.4-0.3%. What is the resolution when the CC1/2 values of the outer shell is 50%? Or the authors should show a Table of CC1/2 for each resolution shell.

5. Similar question as 4. In Supplementary Table 3, The CTX-M-15 drop-on-drop data are refined at a resolution of about 1.6 Å. The CC1/2 values of the outer shell are 10.3-2.2%. What is the resolution of the outer shell when its CC1/2 values are 50%? I would like to see a table of CC1/2 values for each resolution shell.

Answer to points 4 and 5:

The data was merged using program *cxi.merge* from the *cctbx.xfel* package. The resolution cut-offs for the final datasets merged with *cxi.merge* are determined in a standard procedure based on a combination of several criteria, including where the data falls below ten-fold multiplicity, where CC1/2 no longer decreases monotonically and where the values of $I/\sigma(I)$ do not uniformly decrease any more (Nat Methods. 2017 Apr; 14(4): 443–449; Proc Natl Acad Sci U S A. 2020 Jun 9;117(23):12624-12635 or Proc Natl Acad Sci U S A. 2020 Jan 7; 117(1): 300–307). To enable easier inspection of merging results, we included output merging statistics from *cxi.merge* for each resolution shell (Tables 1-9 below). The low CC1/2 in the highest resolution shells is a consequence of how image data is integrated in the processing pipeline (per-image $I/\sigma(I)$ cutoff). If the reviewer and/or editor consider it appropriate, we can incorporate the nine tables into the supplemental information.

The validity of cutoff selection criteria can be confirmed by ‘paired refinement’. Similar to our other studies (Proc Natl Acad Sci U S A. 2020 Jun 9;117(23):12624-12635 or Proc Natl Acad Sci U S A. 2020 Jan 7; 117(1): 300–307), we confronted our applied resolution cutoffs by applying a ‘paired refinement’ procedure to see whether including data from the highest resolution shells has a positive effect on the refinement as compared to more conservative cutoffs. This test was done using the PDB_REDO platform, which includes the implementation of the original ‘paired refinement’ algorithm from Karplus and Diederichs (Science. 2012 May 25;336(6084):1030-3, IUCrJ. 2014 May 30;1(Pt 4):213-20). The ‘paired refinement’ test suggested the same resolution cutoffs for four out of nine datasets included in this publication. For the remaining five, the suggested resolution cutoffs were 0.04-0.1 Å lower. Taking into account that there is always a bit of variability because of the way binning is done, we conclude that the results of ‘paired refinement’ support selection of our resolution cutoffs. Interestingly, for the SACLA HEWL 0.6 s dataset (which has the lowest, CC1/2 value in the highest resolution shell of 0.3%) ‘paired refinement’ gives the same resolution cutoff as the current cutoff. Reversely, for the CTX-M-15 resting state structure from I24 (which CC1/2 value of 56.9% in the highest resolution shell), ‘paired refinement’ suggests slightly lower resolution cutoff (1.71 Å instead of 1.65 Å). This is not intuitive and therefore, the assumption that low CC1/2 in the high resolution shells equates to low quality data and that high CC1/2 automatically means higher data quality are not valid. The benefits of merging protocols applied by cctbx protocols can be demonstrated by electron density map analysis (Proc Natl Acad Sci U S A. 2020 Jun 9;117(23):12624-12635), where it becomes clear that including “weak” data has no perceptible negative influence on the map quality and can in fact help pick out important subtle features in the maps.

We updated the “Data processing” section in “Online methods” by adding the additional information about the applied resolution cutoff criteria and ‘paired refinement’ (P15, lines 367-372 and P16, lines 392-393 in the main manuscript file). In the revised version, Supplementary Tables 2 and 3 are renumbered to Supplementary Tables 5 and 6.

6. In supplementary Table 4, the *k_{cat}* value seems to be quite low. What is the *k_{cat}* in the solution state, not in microcrystals? What is the optimal pH of the enzyme?

CTX-M-15 is known to only poorly hydrolyse carbapenem antibiotics, like ertapenem. It is a well-known property of CTX-M-15 and similar B-lactamase enzymes. This is due to the fact

that while ertapenem (and other similar carbapenem antibiotics) can bind and form a covalent attachment to the active site serine, it only very slowly resolves to the active enzyme (i.e. the steps in panel **b** in Supplementary Figure 12 are very slow). This is reflected in our kinetic data which show a very fast onset of acylation (k_2/K , i.e. the covalent attachment of ertapenem to the active site serine) but a μM inhibition constant ($K_{i\text{app}} = 1.8 \mu\text{M}$). We detail this in the Supplementary Notes and have now added a note to the bottom of Supplementary Table 5 (Supplementary Table 8 in the revised version, P20 in the revised Supplementary Information) to direct readers to the appropriate discussion. Revised Supplementary Table 8 shows the solution state kinetics (not in crystallo kinetics) of CTX-M-15 at pH 7.5. We have altered the heading of revised Supplementary Table 8 to more closely reflect this fact. This pH approximately represents the optimal pH of the enzyme for kinetics, as we note with nitrocefin (a β -lactam reporter substrate) the K_{cat} is $>300 \text{ s}^{-1}$ (Tooke et al, ref. 29 in the online methods references). We have now edited the methods to more explicitly state this, rather than only referring to previous papers.

Tables 1-9 are to be included with response to Referee #1 and #3.

Table 1. Merging statistics for the HEWL resting state structure from SACLA.

Bin	Bin resolution range	Completeness (%)	$CC_{1/2}$	R split	No. obs. multi	$\langle I \rangle$	$\langle I \rangle / \sigma \langle I \rangle$
1	55.72-3.9359	99.75	91.10%	16.60%	195.53	104473	374.75
2	3.9359-3.1239	100	92.40%	15.60%	139.56	99993	318.48
3	3.1239-2.729	100	90.70%	18.10%	118.86	48382	156.515
4	2.729-2.4795	100	89.00%	20.30%	107.05	28741	98.597
5	2.4795-2.3017	100	71.40%	22.30%	98.74	19490	70.897
6	2.3017-2.166	100	87.20%	21.40%	93.06	14893	54.689
7	2.166-2.0575	100	88.10%	20.50%	87.63	10679	40.08
8	2.0575-1.968	100	87.30%	22.20%	83.94	7187	28.168
9	1.968-1.8922	100	85.70%	23.50%	80.43	4848	19.698
10	1.8922-1.8269	100	87.90%	24.10%	77.27	3411	14.463
11	1.8269-1.7698	100	86.20%	25.70%	74.43	2278	10.059
12	1.7698-1.7192	100	85.20%	27.50%	72.03	1827	8.109
13	1.7192-1.6739	100	82.90%	32.80%	69.84	1364	5.982
14	1.6739-1.6331	100	72.70%	43.50%	67.57	1091	4.853
15	1.6331-1.5959	100	61.80%	55.20%	64.68	702	3.273
16	1.5959-1.562	100	32.20%	78.60%	46.99	533	2.2
17	1.562-1.5307	100	23.80%	98.80%	35.30	345	1.26
18	1.5307-1.5018	100	9.30%	112.90%	27.66	290	0.969
19	1.5018-1.475	100	1.20%	124.40%	21.90	251	0.758
20	1.475-1.450	100	5.40%	130.00%	17.08	241	0.583

total	99.99	95.20%	18.70%	80.15	18423	63.741
-------	-------	--------	--------	-------	-------	--------

Table 2. Merging statistics for the HEWL 0.2 s mixing time point structure from SACLA.

Bin	Bin resolution range	Completeness (%)	CC _{1/2}	R split	No. obs multi	<l>	<l>/σ<l>
1	55.72-3.9359	99.75	88.40%	18.90%	130.99	107659	323.747
2	3.9359-3.1239	100	91.40%	16.70%	92.11	104644	277.179
3	3.1239-2.729	100	90.20%	18.70%	78.4	50274	133.907
4	2.729-2.4795	100	87.60%	21.00%	70.08	31330	87.116
5	2.4795-2.3017	100	88.60%	20.40%	64.75	21438	62.498
6	2.3017-2.166	100	91.00%	19.70%	61.13	16616	49.018
7	2.166-2.0575	100	87.10%	22.30%	58	11808	36.419
8	2.0575-1.968	100	84.90%	23.80%	55.11	8042	25.758
9	1.968-1.8922	100	88.30%	22.90%	52.66	5483	18.288
10	1.8922-1.8269	100	83.10%	26.50%	50.91	3689	13.104
11	1.8269-1.7698	100	83.30%	28.50%	48.62	2578	9.317
12	1.7698-1.7192	100	83.50%	30.10%	47.5	2114	7.645
13	1.7192-1.6739	100	80.20%	34.50%	45.41	1530	5.51
14	1.6739-1.6331	100	66.60%	47.70%	44.21	1232	4.579
15	1.6331-1.5959	100	57.30%	58.80%	42.6	828	3.162
16	1.5959-1.562	100	32.20%	83.50%	31.18	615	2.074
17	1.562-1.5307	100	22.20%	100.70%	23.18	450	1.328
18	1.5307-1.5018	100	10.10%	116.40%	18.21	342	0.933
19	1.5018-1.475	100	1.20%	128.00%	14.57	288	0.684
20	1.475-1.45	100	3.30%	129.40%	11.59	245	0.497
total		99.99	94.40%	20.10%	52.84	19464	55.75

Table 3. Merging statistics for the HEWL 0.6 s mixing time point structure from SACLA.

Bin	Bin resolution range	Completeness (%)	CC _{1/2}	R split	No. obs multi	<l>	<l>/σ<l>
1	55.72-3.9359	99.75	89.50%	18.00%	167.18	110881	355.966
2	3.9359-3.1239	100	92.30%	16.40%	118.68	100305	286.125
3	3.1239-2.729	100	90.80%	18.70%	100.43	47194	135.208

4	2.729-2.4795	100	89.10%	20.00%	89.56	28062	85.533
5	2.4795-2.3017	100	88.00%	21.50%	83.55	19140	60.956
6	2.3017-2.166	100	86.00%	21.00%	78.69	14933	47.985
7	2.166-2.0575	100	86.10%	23.00%	73.92	10526	34.344
8	2.0575-1.968	100	86.70%	23.20%	71.02	7097	25.079
9	1.968-1.8922	100	83.90%	24.70%	67.49	4842	17.549
10	1.8922-1.8269	100	83.40%	26.30%	65.08	3307	12.596
11	1.8269-1.7698	100	82.10%	29.50%	61.95	2234	8.775
12	1.7698-1.7192	100	85.30%	30.10%	60.43	1837	7.264
13	1.7192-1.6739	100	77.80%	36.10%	57.76	1372	5.449
14	1.6739-1.6331	100	66.20%	49.90%	55.83	1103	4.362
15	1.6331-1.5959	100	54.40%	62.10%	53.31	710	2.999
16	1.5959-1.562	100	31.80%	84.10%	38.36	526	1.94
17	1.562-1.5307	100	11.00%	107.00%	29.22	338	1.148
18	1.5307-1.5018	100	6.90%	119.50%	22.83	330	0.955
19	1.5018-1.475	100	11.20%	126.70%	17.87	256	0.679
20	1.475-1.45	100	0.30%	133.50%	13.99	266	0.601
total		99.99	94.90%	19.70%	67.39	18671	57.592

Table 4. Merging statistics for the HEWL 2 s mixing time point structure from SACLA.

Bin	Bin resolution range	Completeness (%)	CC _{1/2}	R split	No. obs multi	<l>	<l>/σ<l>
1	55.72-3.9359	99.75	96.30%	11.40%	302.15	117350	437.661
2	3.9359-3.1239	100	96.10%	11.80%	217.59	97487	332.408
3	3.1239-2.729	100	94.70%	13.70%	187.67	43099	154.815
4	2.729-2.4795	100	94.50%	14.10%	169.05	26361	99.23
5	2.4795-2.3017	100	92.80%	14.70%	157.04	18627	73.404
6	2.3017-2.166	100	93.40%	15.70%	149.42	14000	56.67
7	2.166-2.0575	100	94.00%	15.30%	140.02	10203	42.263
8	2.0575-1.968	100	91.90%	16.30%	134.22	7040	30.674
9	1.968-1.8922	100	91.30%	18.30%	128.16	4806	21.675
10	1.8922-1.8269	100	92.00%	18.40%	123.42	3350	15.804

11	1.8269-1.7698	100	90.80%	21.30%	118.23	2331	11.21
12	1.7698-1.7192	100	88.70%	22.40%	115	1892	9.238
13	1.7192-1.6739	100	87.10%	26.60%	107.42	1396	6.776
14	1.6739-1.6331	100	76.10%	40.20%	103.81	1027	5.121
15	1.6331-1.5959	100	71.70%	47.40%	99.4	736	3.737
16	1.5959-1.562	100	47.40%	68.60%	70.01	544	2.403
17	1.562-1.5307	100	23.60%	94.80%	52.27	363	1.415
18	1.5307-1.5018	100	20.20%	97.20%	39.91	368	1.287
19	1.5018-1.475	100	6.90%	118.60%	31.62	291	0.907
20	1.475-1.45	100	3.40%	122.80%	24.65	238	0.648
total		99.99	97.80%	13.60%	125.38	18509	68.773

Table 5. Merging statistics for the CTX-M-15 resting state structure from SACLA.

Bin	Bin resolution range	Completeness	CC _{1/2}	R split	No. obs multi		/σ
1	58.79-4.3431	99.72	81.30%	31.90%	53.39	137696	105.761
2	4.3431-3.4471	100	78.10%	35.30%	41.19	132186	88.37
3	3.4471-3.0113	100	73.50%	41.90%	36.44	71890	47.366
4	3.0113-2.736	100	61.10%	47.60%	33.19	41749	26.957
5	2.736-2.5398	100	49.20%	56.20%	31.02	27810	19.065
6	2.5398-2.3901	100	49.70%	57.00%	29.4	21095	14.992
7	2.3901-2.2704	100	48.90%	58.40%	27.81	15654	11.462
8	2.2704-2.1715	100	36.00%	63.10%	26.93	13271	9.43
9	2.1715-2.0879	100	40.00%	63.90%	25.59	10181	7.492
10	2.0879-2.0159	100	34.60%	67.70%	24.52	7604	5.647
11	2.0159-1.9528	100	31.20%	71.20%	24	6006	4.584
12	1.9528-1.897	100	29.70%	78.50%	22.64	4516	3.486
13	1.897-1.8471	100	36.30%	76.80%	21.88	3355	2.682
14	1.8471-1.802	100	31.10%	81.90%	20.93	2522	2.024
15	1.802-1.761	100	32.40%	89.60%	20.54	1906	1.576
16	1.761-1.7235	100	27.90%	92.80%	20.03	1657	1.328
17	1.7235-1.6891	100	17.40%	104.00%	17.87	1439	1.068
18	1.6891-1.6572	100	12.80%	106.70%	16.13	1515	0.996

19	1.6572-1.6276	100	4.80%	123.60%	15.81	1439	0.952
20	1.6276-1.6	100	2.20%	137.80%	15.16	631	0.447
total		99.98	83.30%	44.10%	26.45	26161	18.484

Table 6. Merging statistics for the CTX-M-15 0.6 s mixing time point structure from SACLA.

Bin	Bin resolution range	Completeness	CC _{1/2}	R split	No. obs multi	<l>	<l>/σ<l>
1	58.83-4.2073	99.74	93.80%	17.40%	231.54	141139	218.829
2	4.2073-3.3394	100	93.10%	18.30%	179.11	123289	169.802
3	3.3394-2.9172	100	87.40%	24.50%	159.44	66625	87.54
4	2.9172-2.6505	100	82.80%	27.20%	145.85	37247	50.529
5	2.6505-2.4605	100	79.50%	29.20%	136.98	25442	37.366
6	2.4605-2.3154	100	83.70%	30.50%	130.28	19749	29.539
7	2.3154-2.1994	100	75.10%	31.50%	122.75	14923	22.753
8	2.1994-2.1037	100	73.50%	32.90%	118.79	12085	18.738
9	2.1037-2.0227	100	74.30%	33.40%	113.26	9100	14.289
10	2.0227-1.9529	100	70.10%	36.60%	109.6	6612	10.802
11	1.9529-1.8918	100	66.50%	40.10%	104.42	5031	8.299
12	1.8918-1.8377	100	67.90%	42.40%	101.4	3506	5.946
13	1.8377-1.7893	100	64.60%	44.50%	97.29	2752	4.743
14	1.7893-1.7457	100	60.30%	49.90%	94.57	1969	3.427
15	1.7457-1.706	100	61.10%	52.90%	90.24	1756	2.965
16	1.706-1.6697	100	56.40%	56.00%	77.96	1627	2.425
17	1.6697-1.6363	100	41.50%	68.70%	74.69	1762	2.611
18	1.6363-1.6054	100	28.80%	103.60%	72.03	713	1.098
19	1.6054-1.5767	100	18.70%	107.70%	59.87	677	0.956
20	1.5767-1.55	100	10.30%	116.30%	43.49	675	0.856
total		99.99	94.90%	23.70%	114.17	24770	36.063

Table 7. Merging statistics for the CTX-M-15 2 s mixing time point structure from SACLA.

Bin	Bin resolution range	Completeness	CC _{1/2}	R split	No. obs multi	<l>	<l>/σ<l>
1	58.78-4.2073	99.74	95.70%	14.40%	271.24	145906	250.768
2	4.2073-3.3394	100	94.00%	16.90%	212.21	121624	188.413
3	3.3394-2.9172	100	90.10%	21.40%	189.04	62875	93.807
4	2.9172-2.6505	100	85.00%	24.80%	174.31	34938	54.358
5	2.6505-2.4605	100	82.90%	27.20%	163.94	23566	39.301
6	2.4605-2.3154	100	81.00%	29.10%	155.03	18082	31.165
7	2.3154-2.1994	100	80.20%	28.70%	147.87	13276	23.471
8	2.1994-2.1037	100	75.30%	30.60%	142.28	10963	19.916
9	2.1037-2.0227	100	74.70%	32.10%	135.73	8057	14.933
10	2.0227-1.9529	100	69.10%	35.10%	131.03	5976	11.266
11	1.9529-1.8918	100	70.20%	36.80%	122.79	4443	8.467
12	1.8918-1.8377	100	73.00%	38.70%	118.52	3103	6.061
13	1.8377-1.7893	100	66.30%	43.60%	113.74	2318	4.589
14	1.7893-1.7457	100	67.20%	45.60%	110.5	1708	3.378
15	1.7457-1.706	100	56.20%	53.50%	104.86	1465	2.822
16	1.706-1.6697	100	52.10%	58.00%	88.36	1421	2.401
17	1.6697-1.6363	100	28.70%	69.90%	83.95	1615	2.637
18	1.6363-1.6054	100	17.90%	110.50%	80.36	508	0.863
19	1.6054-1.5767	100	13.70%	113.40%	67.46	582	0.915
20	1.5767-1.55	100	9.70%	123.20%	48.45	512	0.714
total		99.99	96.10%	21.00%	134.25	24082	39.566

Table 8. Merging statistics for the CTX-M-15 resting state structure from DLS I24.

Bin	Bin resolution range	Completeness	CC _{1/2}	R split	No. obs multi	<l>	<l>/σ<l>
1	59.24-4.4788	100	79.80%	29.50%	177.17	96951	169.219
2	4.4788-3.5548	100	86.20%	25.50%	123.16	124327	162.179
3	3.5548-3.1054	100	90.90%	22.40%	105.76	83039	112.918
4	3.1054-2.8215	100	88.40%	20.90%	94.38	50913	78.035
5	2.8215-2.6192	100	88.50%	21.50%	81.8	35555	57.477

6	2.6192-2.4648	100	89.20%	22.90%	76.73	29081	51.254
7	2.4648-2.3413	100	86.10%	23.20%	68.38	25947	45.527
8	2.3413-2.2394	100	85.30%	24.20%	65.53	20755	39.15
9	2.2394-2.1532	100	86.20%	23.10%	58.95	18950	35.135
10	2.1532-2.0789	100	84.70%	24.30%	55.74	16067	31.518
11	2.0789-2.0139	100	82.70%	25.40%	50.78	13091	27.13
12	2.0139-1.9563	100	81.60%	26.90%	45.25	10998	23.247
13	1.9563-1.9048	100	85.30%	25.80%	41.47	9564	20.773
14	1.9048-1.8583	100	79.80%	29.80%	35.68	7443	16.986
15	1.8583-1.8161	100	80.50%	30.00%	31.44	6083	13.958
16	1.8161-1.7774	100	78.20%	32.20%	26.86	5258	12.242
17	1.7774-1.7418	100	72.20%	36.20%	22.92	4399	10.395
18	1.7418-1.709	100	72.40%	39.00%	17.79	3978	8.879
19	1.709-1.6785	100	61.70%	42.60%	14.19	3709	7.909
20	1.6785-1.65	99.93	56.90%	48.80%	11.15	3402	6.376
total		100	89.30%	25.40%	61.27	29215	47.659

Table 9. Merging statistics for the CTX-M-15 10 min time point from DLS I24.

Bin	Bin resolution range	Completeness	CC _{1/2}	R split	No. obs multi		/σ
1	59.52-4.4788	100	84.60%	27.10%	135.16	107616	200.233
2	4.4788-3.5548	100	86.90%	22.80%	91.72	135888	192.85
3	3.5548-3.1054	100	87.70%	21.10%	79.06	84145	125.937
4	3.1054-2.8215	100	88.60%	21.10%	69.27	51481	85.263
5	2.8215-2.6192	100	87.60%	21.40%	62	36435	65.238
6	2.6192-2.4648	100	88.70%	23.00%	57.64	31399	57.635
7	2.4648-2.3413	100	89.40%	21.90%	52.07	26078	49.277
8	2.3413-2.2394	100	87.30%	22.60%	49.63	21770	43.712
9	2.2394-2.1532	100	83.70%	24.60%	45.1	18994	37.154
10	2.1532-2.0789	100	85.20%	24.90%	43.58	16509	34.519
11	2.0789-2.0139	100	81.60%	28.20%	39.41	13351	28.661
12	2.0139-1.9563	100	81.30%	27.40%	36.33	11190	24.585
13	1.9563-1.9048	100	83.20%	27.20%	33.95	9227	21.612

14	1.9048-1.8583	100	80.70%	29.80%	28.81	7622	17.54
15	1.8583-1.8161	100	77.30%	31.20%	26.35	6379	15.229
16	1.8161-1.7774	100	72.80%	34.40%	22.57	5064	12.461
17	1.7774-1.7418	100	77.50%	36.20%	18.6	4682	10.968
18	1.7418-1.709	100	71.10%	41.10%	14.38	4092	8.938
19	1.709-1.6785	100	58.20%	47.20%	11.6	3511	7.32
20	1.6785-1.65	99.93	25.80%	58.30%	9.2	3356	6.4
							
total		100	90.20%	24.40%	47.09	30765	53.658

REVIEWER COMMENTS

Reviewer #1 (Remarks to the Author):

The authors have addressed all points raised in the first review.
I support the publication of this manuscript in its current form.

Reviewer #2 (Remarks to the Author):

The authors have thoroughly answered my comments in detail, clarifying the points I have made, improving the manuscript. Therefore I recommend this article for publication.

Reviewer #3 (Remarks to the Author):

The authors have sufficiently responded to my questions/comments and the manuscript has greatly improved.
The Peer Review File will be open to the public and the author's claims about data processing conveyed to the readers including the Tables of merging statistics, so there is no problem.
I think it is ready for publication.

Eiichi Mizohata